# Differential conformational dynamics in two type-A RNA-binding domains drive the double-stranded RNA recognition and binding

Firdousi Parvez[1], Devika Sangpal[2], Harshad Paithankar[3], Zainab Amin[3], Jeetender Chugh[3]*

[1]Department of Biology, Indian Institute of Science Education and Research (IISER), Pune, India; [2]Department of Biotechnology (with jointly merged Institute of Bioinformatics and Biotechnology), Savitribai Phule Pune University, Pune, India; [3]Department of Chemistry, Indian Institute of Science Education and Research (IISER), Pune, India

*For correspondence:
cjeet@iiserpune.ac.in

Competing interest: The authors declare that no competing interests exist.

## Abstract

Trans-activation response (TAR) RNA-binding protein (TRBP) has emerged as a key player in the RNA interference pathway, wherein it binds to different pre-microRNAs (miRNAs) and small interfering RNAs (siRNAs), each varying in sequence and/or structure. We hypothesize that TRBP displays dynamic adaptability to accommodate heterogeneity in target RNA structures. Thus, it is crucial to ascertain the role of intrinsic and RNA-induced protein dynamics in RNA recognition and binding. We have previously elucidated the role of intrinsic and RNA-induced conformational exchange in the double-stranded RNA-binding domain 1 (dsRBD1) of TRBP in shape-dependent RNA recognition. The current study delves into the intrinsic and RNA-induced conformational dynamics of the TRBP-dsRBD2 and then compares it with the dsRBD1 study carried out previously. Remarkably, the two domains exhibit differential binding affinity to a 12-bp dsRNA owing to the presence of critical residues and structural plasticity. Furthermore, we report that dsRBD2 depicts constrained conformational plasticity when compared to dsRBD1. Although, in the presence of RNA, dsRBD2 undergoes induced conformational exchange within the designated RNA-binding regions and other residues, the amplitude of the motions remains modest when compared to those observed in dsRBD1. We propose a dynamics-driven model of the two tandem domains of TRBP, substantiating their contributions to the versatility of dsRNA recognition and binding.

## eLife assessment

This study presents a **useful** comparison of the dynamic properties of two RNA-binding domains. The data collection and analysis are **solid**, making excellent use of a suite of NMR experiments and ITC data. Nonetheless, reported evidence was found to only partially support the proposed connection between the backbone dynamics of the tandem domains and their RNA binding activity. This work will be of interest to biophysicists working on RNA-binding proteins.

## Introduction

RNA-binding proteins (RBPs) play a crucial role in every aspect of RNA biology, including folding, splicing, processing, transport, and localization (*Dreyfuss et al., 2002*; *Fu et al., 2016*; *Jiang and Baltimore, 2016*; *Kim et al., 2009*; *Wilkinson and Shyu, 2001*). RBPs can be broadly classified into

single-stranded RNA-binding proteins (ssRBPs) and double-stranded RNA-binding proteins (dsRBPs). While ssRBPs are generally sequence specific as the interaction points include exposed nucleobases (*Auweter et al., 2006*; *Daubner et al., 2013*); dsRBPs tend to target a particular structural fold, especially the A-form RNA duplex via phosphate backbone and sugar hydroxyl protons (*Masliah et al., 2013*). dsRBPs interact with highly structured dsRNAs via their double-stranded RNA-binding motifs/domains (dsRBMs/dsRBDs).

dsRBDs are 65–68 amino acid long domains (*St Johnston et al., 1992*) that contain a highly conserved secondary structure ($\alpha_1$–$\beta_1$–$\beta_2$–$\beta_3$–$\alpha_2$), where the two α-helices are packed against an antiparallel β-sheet formed by three β-strands (*Bycroft et al., 1995*; *Kharrat et al., 1995*). All the dsRBDs have three common RNA-binding regions: (1) the middle of helix $\alpha_1$ (E residue), (2) the N-terminal residues of helix $\alpha_2$ (KKxAK motif), and (3) the loop between $\beta_1$ and $\beta_2$ strands (GPxH motif), forming a canonical RNA-binding surface that interacts with the consecutive minor, major, and minor groove of a dsRNA (*Masliah et al., 2013*; *Tian et al., 2004*). dsRBDs interact with dsRNAs in a non-sequence-specific manner, targeting the sugar 2'-OH groups (dsRBDs do not target dsDNAs) and phosphate backbone (*Bevilacqua and Cech, 1996*). They are often arranged in a modular fashion to create highly versatile RNA-binding surfaces and have been classified into types A and B (*Fierro-Monti and Mathews, 2000*; *St Johnston et al., 1992*; *Krovat and Jantsch, 1996*; *Masliah et al., 2013*). Type-A dsRBDs are highly homologous (>59%) to the consensus sequence and are involved in RNA binding, whereas type B has only a conserved C-terminal end and helps in stabilizing the protein–RNA/protein complex (*Green et al., 1995*; *Krovat and Jantsch, 1996*; *Laraki et al., 2008*).

dsRBDs, both within and across proteins, exhibit significant variations in their binding modes and affinities toward specific RNA targets. For example, *X. laevis* RNA-binding protein A (XlrbpA) has three dsRBDs: xl1 (type A), xl2 (type A), and xl3 (type B), where the xl2 is only able to bind the dsRNA in vitro (*Krovat and Jantsch, 1996*). In Protein Kinase R (PKR), these two types of dsRBDs also demonstrate a differential dynamic behavior, i.e., dsRBD1 (type A) shows plasticity on μs and ps–ns timescale, while dsRBD2 (type B) predominantly depicts dynamics on the ps–ns timescale dynamics (*Fierro-Monti and Mathews, 2000*; *Nanduri et al., 2000*). Furthermore, the average order parameter ($S^2$; the degree of motion of the backbone N–H vector in a cone, where $S^2 = 1$ indicates limited flexibility and 0 indicates maximum flexibility) for the individual α-helices and β-sheets is observed to be lower for dsRBD1 than for dsRBD2, suggesting the latter as a more rigid domain (*Fierro-Monti and Mathews, 2000*; *Nanduri et al., 2000*). Trans-activation response (TAR) RNA-binding protein (TRBP) harbors two type A (*Krovat and Jantsch, 1996*) dsRBDs, i.e., dsRBD1 and dsRBD2, where the latter shows a significantly stronger binding affinity (four times) for pre-miR-155 (*Benoit et al., 2013*). Additionally, studies have reported that dsRBD2 of TRBP displays a higher binding affinity for siRNA and HIV TAR RNA (*Daviet et al., 2000*; *Yamashita et al., 2011*). It is intriguing to notice that the two type-A dsRBDs originating from the same protein demonstrate a markedly different binding affinity for target RNAs across various cases, as elucidated above. Recent investigations have suggested that the dissimilar behavior of dsRBDs in Dicer (*Wostenberg et al., 2012*), PKR (*Nanduri et al., 2000*), and double-stranded-RNA-binding protein 4 (DRB4) (*Chiliveri et al., 2017*) might be attributed to protein dynamics. Although variations in the binding affinity for the two type-A dsRBDs of TRBP have been documented for several target RNAs, a comprehensive exploration of the differences in the conformational dynamics between these two domains remains to be undertaken.

In this study, we have investigated the role of differential conformational dynamics of the two type-A dsRBD domains of TRBP2 (isoform 1 of TRBP) in RNA recognition and binding. First discovered as a TRBP (39 kDa) involved in HIV-I replication (*Gatignol et al., 1991*), TRBP was later found to be indispensable to the RNA interference (RNAi) pathway (*Kim et al., 2014*). It is involved in Dicer-mediated pre-miRNA/pre-siRNA cleavage and recruitment of Argonaute protein (*Chendrimada et al., 2005*). Being a part of the RNA-induced silencing complex (RISC), TRBP helps in the guide strand selection (*Noland et al., 2011*; *Noland and Doudna, 2013*). The guide strand of the mature miRNA directly interacts with the target mRNA to regulate its expression, while the passenger strand is cleaved off by the RISC (*Leuschner et al., 2006*; *Matranga et al., 2005*). TRBP and its homologs across different species such as Loquacious (Loqs in *D. melanogaster*), R2D2 (*D. melanogaster*), DRB1-3,5 (*A. thaliana*), RNAi defective 4 (RDE-4 in *C. elegans*), Xlrbpa (*X. laevis*) (*Eckmann and Jantsch, 1997*), and the PKR activator (PACT) protein (mammals) (*Peters et al., 2001*) have conserved arrangement of three consecutive dsRBDs; of which, the two N-terminal domains (type A) are known to

bind dsRNAs, whereas the third domain is known for protein–protein interactions. We have recently established the role of TRBP2-dsRBD1 dynamics in dsRBD–dsRNA interactions and proposed that dsRBD1 adopts a conformationally dynamic structure to recognize a set of topologically different dsRNA structures (*Paithankar et al., 2022*). The µs timescale motions were found to be present all along the dsRBD1 backbone with higher frequency motions ($k_{ex} > 50$ kHz) in the RNA-binding sites. The data suggested that the presence of conformational exchange in the µs timescale could help the dsRBD1 to dynamically tune itself for targeting conformationally distinct dsRNA substrates.

In this work, we have compared TRBP2-dsRBD1 with TRBP2-dsRBD2 in terms of the structure, dynamics (intrinsic and RNA-induced), and dsRBD–dsRNA interactions in the two type-A domains. We have measured motions in the TRBP2-dsRBD2 at ps–ns and µs–ms timescale dynamics by Nuclear Magnetic Resonance (NMR) relaxation dispersion experiments in apo-state and studied its perturbation in the presence of an A-form duplex RNA. We also compared the apo- and RNA-bound conformational dynamics measured in dsRBD2 with that of dsRBD1. Based on our observations, we propose that the differential protein dynamics and its perturbation in the presence of RNA in the two dsRBDs enables them to recognize a variety of RNA substrates and may lead to diffusion along the length of the RNA.

## Results and discussion
### Structural and dynamical comparison of apo TRBP2-dsRBD1 and TRBP2-dsRBD2

Primary sequence comparison of TRBP2-dsRBD1 and TRBP2-dsRBD2 domain constructs (*Figure 1A*) revealed 30% identity and 38% similarity. The consensus for dsRNA-binding has been marked in red in *Figure 1B*. The two reported RNA-binding regions 1 (E) and 3 (KKxAK) of both the domains were matched to the consensus (*Figure 1B*). While RNA-binding region 2 (GPxH) (*Figure 1B*, region 2) of TRBP2-dsRBD2 was an exact match to the consensus, dsRBD1 harbored a mutation in region 2 (P56Q). Proline is a rigid amino acid with one less dihedral angle; it imparts flexibility to the backbone by causing secondary structure breaks (*Imai and Mitaku, 2005*; *Krieger et al., 2005*). Thus, the presence of conserved Pro186 in the $\beta_1$–$\beta_2$ loop of dsRBD2 (*Figure 1B*, region 2) may perturb the $\beta_1$–$\beta_2$ loop region plasticity, thereby making it more accessible to the incoming RNA partner. Additionally, dsRBD2 contains a KR-helix motif in the $\alpha_2$-helix (*Figure 1B*, region 3) known to increase its binding affinity, as reported earlier by *Daviet et al., 2000*. Owing to these tightly conserved RNA-binding regions and the presence of additional KR-helix motif, dsRBD2 could make stronger contact with RNA.

The size-exclusion chromatography coupled with multiple angle light scattering (SEC-MALS) study of TRBP2-dsRBD2 in the experimental conditions used showed a single monomeric species in solution (*Figure 1—figure supplement 1*). The $^1$H-$^{15}$N HSQC spectrum (*Figure 1—figure supplement 2*) of TRBP2-dsRBD2 indicated a well-folded protein. It (154–234 aa) was compared to the previously assigned spectrum of the TRBP2-D1D2 construct (19–228 aa) (*Benoit and Plevin, 2013*) to transfer the backbone amide resonance assignments. Resonance assignments were further confirmed using a set of double and triple resonance experiments, and a total of 73 non-proline residues were assigned, as shown in the representative $^1$H-$^{15}$N HSQC (*Figure 1—figure supplement 2*). Overall, 89% $^1$H (151/169), 74% $^{13}$C (124/168), and 87% $^{15}$N (73/84) resonances from the backbone, and 43% $^1$H (174/402) and 25% $^{13}$C (59/238) from the side chains were assigned. Similar to the CS-Rosetta structure calculated previously for dsRBD1 (*Figure 1C*) by our group (*Paithankar et al., 2018*), the CS-Rosetta structure calculated for the dsRBD2 (*Figure 1D*) matched well with the previously reported structures in terms of characteristic dsRBD fold. The core residues (159–227 aa) of dsRBD2 were found to adapt the characteristic dsRBD αβββα fold with an RMSD (root mean square deviation) of 1.284 Å when compared to the previously reported solution structure of dsRBD2 (2CPN; *Yamashita et al., 2011*). An alignment of core structures between dsRBD1 and dsRBD2 (*Figure 1E*) yielded an RMSD of 0.894 Å, indicating a close match between the two domains. Despite their core length being identical (69 aa), the individual secondary structure spans were found to be different, as listed in *Supplementary file 1a*. The flexible regions, like loops 1 and 2 depicted in *Figure 1B, E*, are longer in dsRBD2, whereas the structured regions $\beta_2$, $\beta_3$, and $\alpha_2$ are longer in dsRBD1. Most importantly, loop 2 ($\beta_1$–$\beta_2$ loop) – critical for RNA binding (*Masliah et al., 2013*) – is equal to the canonical length in dsRBD2, while in dsRBD1, it is shorter by 1 residue (*Figure 1E*). We have used a 12-bp duplex RNA for the comparative

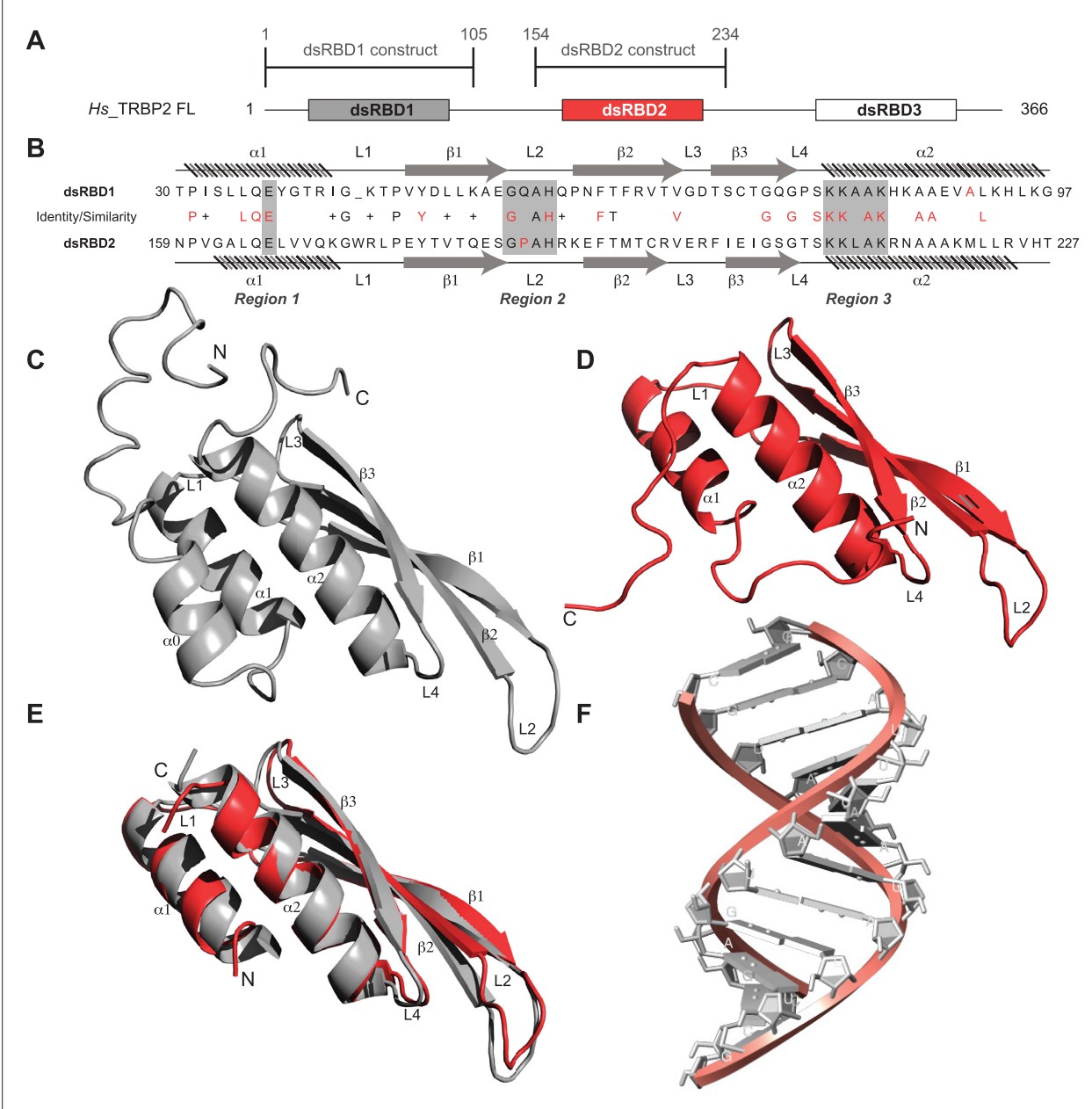

**Figure 1.** Sequence and structure comparisons for dsRBD1 and dsRBD2 of TRBP2 along with the structure of the D12 RNA duplex. (**A**) dsRBD constructs [TRBP2-dsRBD1 (1–105 aa) and TRBP2-dsRBD2 (154–234 aa)] of Human TRBP2 full-length protein used in this study. (**B**) Sequence alignment of the two constructs mentioned in (**A**). CS-Rosetta structures of (**C**) TRBP2-dsRBD1 (*Paithankar et al., 2018*), (**D**) TRBP2-dsRBD2, (**E**) an alignment of core residues of dsRBD1 and dsRBD2, and (**F**) model structure of 12 bp duplex D12 RNA.

The online version of this article includes the following figure supplement(s) for figure 1:

**Figure supplement 1.** Size-exclusion chromatography coupled with multiple angle light scattering (SEC-MALS) elution profile for TRBP2-dsRBD2 showing that the protein remains monomeric in the buffer conditions used for NMR studies.

**Figure supplement 2.** Backbone resonance assignments for TRBP2-dsRBD2 marked on the $^1$H-$^{15}$N HSQC recorded on 600 MHz NMR spectrometer at 298 K in buffer D, pH 6.4.

**Figure supplement 3.** $^1$H-NMR spectrum of the imine region of the annealed D12 RNA indicating the duplex formation.

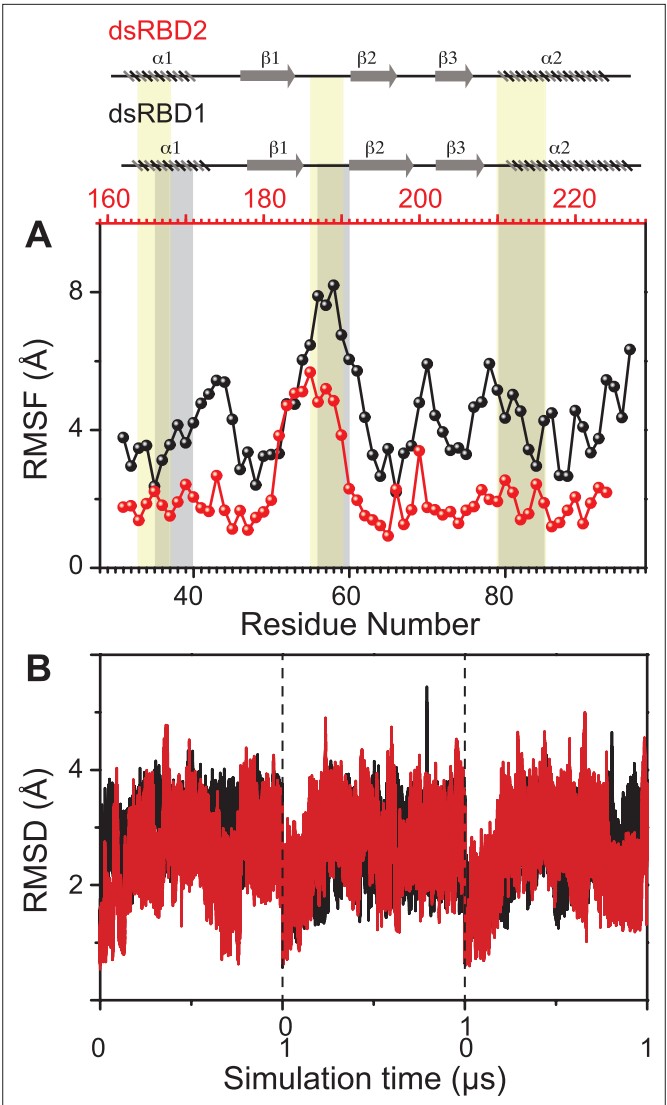

**Figure 2.** Dynamics comparison of dsRBD1 and dsRBD2 as measured by MD simulations. (**A**) RMSF profile (by C$_\alpha$) of core dsRBD1 (black) and dsRBD2 (red). The secondary structure for the two domains has been shown on the top, and three RNA-binding regions in dsRBD1 and dsRBD2 have been highlighted using vertical grey and yellow bars, respectively. (**B**) RMSD of profiles of dsRBD1 (black) and dsRBD2 (red) over 1 µs simulation time. RMSD values for data measured in triplicate have been separated by vertical dashed lines.

study called the D12 RNA derived from the miR-16-1 (*Figure 1F*, *Figure 1—figure supplement 3* and sequence mentioned in *Supplementary file 1b*).

Comparing the molecular dynamics (MD) simulation data of the core residue-specific RMSF (root mean square fluctuation) plots of these two type-A dsRBD domains (*Figure 2A*), we observe that the average RMSF of the core domain of dsRBD2 is 0.22 nm, which is almost half as compared to dsRBD1 RMSF average of 0.43. This, in turn, indicates that dsRBD2 is overall more rigid among the two. The most interesting fact is that loop 2 (shown in *Figure 1B*) of dsRBD2 is very flexible compared to the rest of the core, with an average RMSF of 0.49 nm (more than double the core average), as depicted in *Figure 2A*. Hence, we propose that this long, conserved, and dynamic loop 2 might help dsRBD2 to establish strong contacts in the minor grooves of dsRNA partners. An effect of conserved insertions in the β$_1$–β$_2$ loop on RNA binding has been previously discussed in the crystal structure of full-length *Arabidopsis* HEN1 in a complex with a small RNA duplex (*Huang et al., 2009*). The MD simulation data of RMSD against the simulation time of TRBP2-dsRBD2 showed that the protein remained stable during the course of triplicate simulations, as seen previously in the case of dsRBD1 (*Figure 2B*).

To gain insights into the picosecond to nanosecond (ps–ns) timescale motions of apo-dsRBD2, NMR spin relaxation data ($R_1$, $R_2$, and [$^1$H]-$^{15}$N nOe) were recorded on 600 and 800 MHz NMR spectrometers (**Figure 3**). The set of plots for $R_1$, $R_2$, and [$^1$H]-$^{15}$N nOe at both 600 and 800 MHz magnetic fields showed the expected trend (**Figure 3—figure supplement 1** and **Supplementary file 1c**). $R_1$ in the structured residues showed lower values at a higher magnetic field and similar values in the terminal regions at the two magnetic fields (**Figure 3—figure supplement 1A**). The increase of core $R_1$ rates (decreasing $T_1$) with increasing magnetic field indicates that most of the core residues lie on the right-hand side of the $T_1$ minima in the standard plot of $T_1$ vs. $\omega\tau_c$, where $\omega$ is the frequency of motions, $\tau_c$ and is rotational correlation time (**Bloembergen et al., 1948**). The core residues had average $R_1$ rates of 1.43 ± 0.05 s$^{-1}$ (**Figure 3**). The $R_2$ rates (**Figure 3—figure supplement 1B**) of a few N-terminal (N159) and core residues (A163, V169, R174, E177, V180, R189, M194, R197, V198, G206, G207, G208, K210, L212, 221, 224, and A227) marginally increased (0.1–3 s$^{-1}$) when measured at 800 MHz than when measured at 600 MHz, suggesting a very insignificant presence of the $R_{ex}$ component in the linewidth of these residues in the experimental conditions. The dsRBD2 core N- and C-terminal ends (S151–E157, N159, T227, V228, L230, and A232) and the loop residues (E177, G185, E191, S209, and K210) exhibited relatively higher flexibility as indicated by lower $R_2$ rates in **Figure 3B** and nOe values in **Figure 3C**. A few residues lying in different secondary structures have been shown in **Figure 3B**, for e.g.: Q165, L167, and V168 in $\alpha_1$; T193 and T195 in $\beta_2$; L212, A213, N216, and L223 in $\alpha_2$ exhibited only slightly higher (>1 s$^{-1}$) than the average $R_2$ rate (10.92 ± 0.37 s$^{-1}$), indicating a thin conformational exchange in the secondary structured regions.

The core (159–227 aa) average $R_1$ (dsRBD1 = 1.52 ± 0.03 s$^{-1}$; dsRBD2 = 1.43 ± 0.05 s$^{-1}$) and $R_2$ rates (dsRBD1 = 9.83 ± 0.17 s$^{-1}$; dsRBD2 = 10.92 ± 0.37 s$^{-1}$) for the two domains were very similar (Figure S15 of **Paithankar et al., 2022**). In contrast, dsRBD1 (Figure S15C of **Paithankar et al., 2022**) had significantly lower average heteronuclear steady-state nOe values (0.63 ± 0.02) than dsRBD2 (0.73 ± 0.03) shown in **Figure 3C**, indicating a more rigid core at the picosecond timescale motions in dsRBD2, thereby supporting the earlier observations made in the MD simulation study (**Figure 2**).

The extended model-free analysis of the relaxation data for the two core domains allowed us to compare the order parameters S$^2$, $R_{ex}$ components, and global tumbling time ($\tau_c$). The overall order parameters for dsRBD2 (**Supplementary file 1d**) followed the expected trend of higher values in the secondary structured regions and lower values in the terminals and loops (**Figure 3—figure supplement 2**, top panel). Interestingly, the secondary structural motifs $\alpha_1$, loop 2, and $\alpha_2$ (refer to **Figure 1B, D**), containing RNA-binding regions of dsRBD2, showed a higher rigidity than dsRBD1. The presence of conservedf Pro in loop 2 (L2) of dsRBD2 and a longer length of L2 in dsRBD2 (**Supplementary file 1a**) might be responsible for the additional observed flexibility in dsRBD2, as also observed in the MD simulations (**Figure 2**). Loop 1 (W173, L175) of dsRBD2 containing the stabilizing tryptophan (absent in dsRBD1) also showed higher rigidity than dsRBD1, corroborated by the low RMSF values for residues in this region from the MD simulation study. The global rotational correlation time ($\tau_c$) of dsRBD2 was 7.3 ns, and that of dsRBD1 was 7.64 ns, suggesting a more compact structure of dsRBD2 than dsRBD1. This result corroborates the CD-based determination of the melting point of the two constructs. While TRBP2-dsRBD2 exhibited a higher $T_m$ (55°C), TRBP2-dsRBD1 melted at 45°C (data not shown), thus suggesting a stronger network of interactions in TRBP2-dsRBD2 (**Yamashita et al., 2011**). Indeed, it has been shown previously that the tryptophan (W173) present in the $\alpha_1$–$\beta_1$ of dsRBD (W is absent in dsRBD1 at this position) induced local hydrophobic and cation–π interactions with K171, E199, R200, F201, and V225, thereby enhancing the overall thermal stability of dsRBD2 (**Yamashita et al., 2011**).

Only seven residues of dsRBD2 could fit into a model-free model containing the exchange term ($R_{ex}$) (**Supplementary file 1e**) as opposed to 11 residues in dsRBD1, suggesting that there is significantly lower conformational flexibility at the μs–ms timescale in dsRBD2 shown in **Figure 3—figure supplement 2**, bottom panel. While dsRBD2 depicted a low-frequency (<2 s$^{-1}$) contribution to the line width of resonances in the RNA-binding regions (E166, L212, N216, and L223), dsRBD1 had a larger contribution (2–8 s$^{-1}$) to the line width of resonances present all along the backbone (**Figure 3—figure supplement 2**, bottom panel). In summation, the plasticity at the ps–ns timescale is present in both the dsRBDs of TRBP2, while the amplitude of motions is found to be largely restricted for the dsRBD2.

Similar to dsRBD1, dsRBD2 does not exhibit motions at a slower ms timescale as probed by the Carr–Purcell–Meiboom–Gill (CPMG) relaxation dispersion experiments shown in **Figure 3—figure**

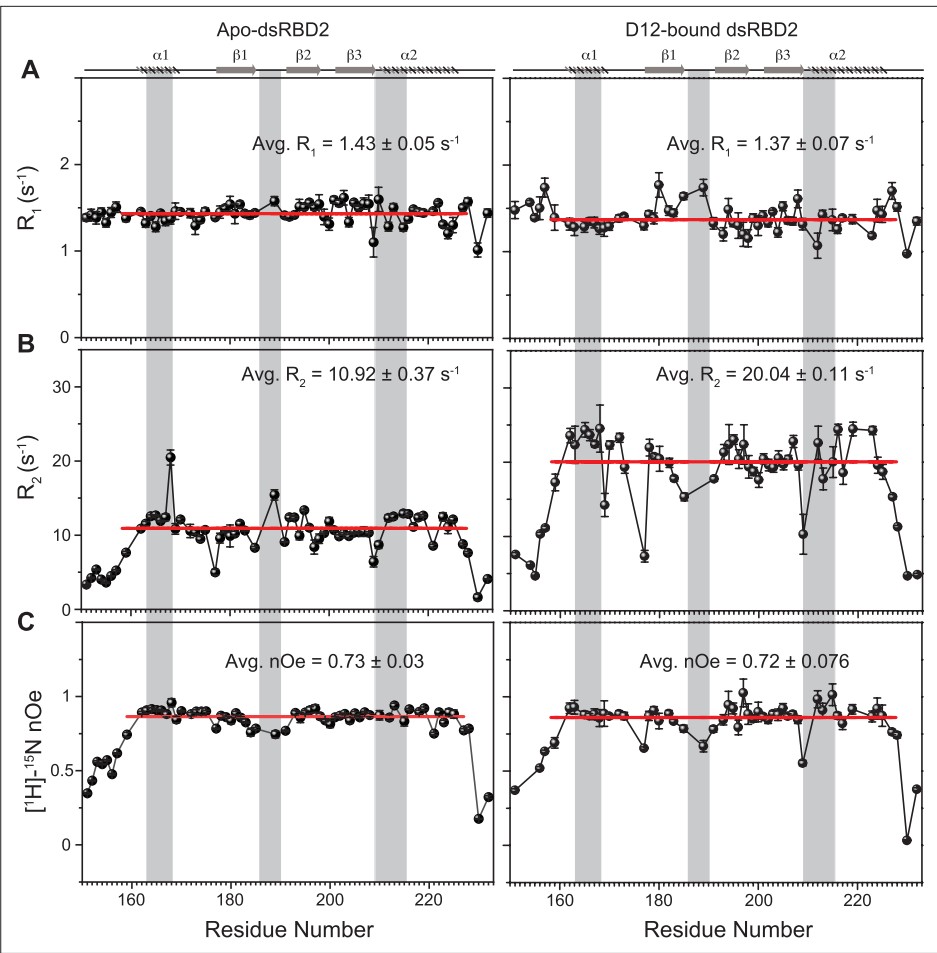

**Figure 3.** Spin relaxation parameters (**A**) $R_1$, (**B**) $R_2$, and (**C**) [$^1$H]-$^{15}$N nOe plotted against the residues for (left panel) apo TRBP2-dsRBD2 and (right panel) D12-bound TRBP2-dsRBD2. Experiments were recorded on a 600-MHz NMR spectrometer at 298 K. The secondary structure of TRBP2-dsRBD2 has been mentioned at the top, and the RNA-binding region of the protein has been marked in grey vertical columns. Average $R_1$, $R_2$, and [$^1$H]-$^{15}$N-nOe of the core residues (159–227 aa) at 600 MHz has been depicted in the green bar, and at 800 MHz has been depicted in the red bars. The average values for spin relaxation parameters were calculated for 68 core residues (159-227 aa), and the standard error of the mean is calculated. The errors in the relaxation rates were calculated using duplicate delay data (two delays in duplicate) and Monte Carlo simulations (n=500). Errors in nOe values were obtained by propagating the errors from root-mean-square deviation (RMSD) values of baseline noise as obtained from Sparky in the respective spectra.

The online version of this article includes the following figure supplement(s) for figure 3:

**Figure supplement 1.** Comparison of fast dynamics (ps-ns timescale) for TRBP2-dsRBD2 as measured on two magnetic field strengths.

**Figure supplement 2.** Top panel: Order parameters ($S^2$), and bottom panel: $R_{ex}$ as calculated using model-free fitting of the fast relaxation data for core residues of TRBP2-dsRBD1 (histograms) and TRBP2-dsRBD2 (red scatter) plotted against residue numbers.

**Figure supplement 3.** $R_{2,eff}$ rates plotted against $\nu_{CPMG}$ for the 65 non-overlapping residues of apo TRBP2-dsRBD2 measured at 600 MHz at 298 K.

**Figure supplement 4.** Top panel: Order parameters ($S^2$), and bottom panel: $R_{ex}$ as calculated using model-free fitting of the fast relaxation data for apo TRBP2-dsRBD2 (left panel) and D12 RNA-bound TRBP2-dsRBD2 (right panel) plotted against residue numbers.

**Figure supplement 5.** $R_{2,eff}$ rates plotted against $\nu_{CPMG}$ for the 63 non-overlapping residues of TRBP2-dsRBD2 measured in the presence of D12 RNA at 600 MHz at 298 K.

supplement 3 and *Supplementary file 1f*. No dispersion was observed in the $R_{2,eff}$ rates with increasing $\nu_{CPMG}$, suggesting the absence of motions sensitive to this experiment (*Paithankar et al., 2022*).

The heteronuclear adiabatic relaxation dispersion (HARD) NMR experiment (*Mangia et al., 2010*; *Traaseth et al., 2012*) was used to study NMR spin relaxation in a rotating frame. The dispersion in relaxation rates is created by changing the shape of a hyperbolic secant (HS$n$, where $n$ = stretching factor) adiabatic pulse that is used to create the spin lock. HARD experiments are sensitive to the conformational exchange processes occurring on the 10 µs to 10 ms timescales. The $R_{1\rho}$ and $R_{2\rho}$ relaxation rates showed that with the increase in applied spin-lock field strength from HS1 to HS8, the $R_{1\rho}$ rates increased (*Figure 4A, C* and *Supplementary file 1g*), and the $R_{2\rho}$ rates decreased (*Figure 4B, D* and *Supplementary file 1h*) for both dsRBD1 and dsRBD2. The extent of dispersion was much less in the core residues of dsRBD2, indicating the absence of higher conformational dynamics in the dsRBD2. The $R_{1\rho}$ and $R_{2\rho}$ rates were then fit to Bloch–McConnell equations by using the geometric approximation approach (*Chao and Byrd, 2016*) to extract the exchange rate constant ($k_{ex}$) and the excited state population ($p_B$) (*Figure 5A* and *Supplementary file 1i*). The number of residues with $k_{ex} > 5000$ Hz and >10% $p_B$ (red and green, medium and big spheres) (*Figure 5C*) differed vastly between the two domains. Only three residues with such conformational exchange in dsRBD2 were identified: R200 (loop 3), L212, and R224 ($\alpha_2$); whereas in dsRBD1, the number is 15, distributed along the entire backbone of the protein (Figure 6 of *Paithankar et al., 2022*), similar to PKR-dsRBD1, where 75% of the residues from dsRBM1 showed $R_{ex} > 1$ s$^{-1}$ (*Nanduri et al., 2000*). Interestingly, all three conserved RNA-binding regions in dsRBD1: $\alpha_1$ (L35, Q36), $\beta_1$–$\beta_2$ loop (E54), and $\alpha_2$ (N61) were brimming with a $k_{ex} > 50,000$ Hz. To sum it up, most of the dsRBD2 core showed slow conformational exchange ($k_{ex} < 5000$ Hz), depicted in blue small spheres, while the dsRBD1 domain was undergoing significantly faster conformational exchange ($k_{ex} > 5000$ Hz). Taken together, the intrinsic $k_{ex}$ profile of TRBP2-dsRBD1 and TRBP2-dsRBD2 revealed the presence of 10 µs to 10 ms timescale conformational dynamics distributed throughout the core dsRBD rather than being localized in the RNA-binding regions; however, the amplitude of motions was found significantly smaller in dsRBD2.

A similar extensive fast and slower timescale conformational dynamics has also been studied in the case of DRB4 protein (*Chiliveri et al., 2017*), where the first domain was found to have a large conformational exchange when compared with that of the second domain. The conformational plasticity of dsRBD1 enables it to initiate primary interaction with various structurally and sequentially different dsRNAs in the TRBP2 protein (*Paithankar et al., 2022*).

## dsRBD2 has a higher affinity for a short double-stranded A-form RNA duplex

Target RNA sequences were designed based on the fact that TRBP interacts with miR-16-1 duplex (miRbase accession no. MI0000070) (*Yan et al., 2019*). Three mutants of miR-16-1 were created to perturb the RNA shapes as described previously (*Paithankar et al., 2022*). Briefly, (1) wt miR-16 (22–23 bp) has a bulge (unpaired uridine) and an internal loop (A:A mismatch), miR-16-1-M has only A:A mismatch, (2) miR-16-1-B has only U-bulge, and (3) miR-16-1-D has neither the bulge nor internal loop forming a perfect duplex (Figure 1 of *Paithankar et al., 2022*). It has been established that the TRBP2-dsRBD1 is able to recognize a set of topologically different dsRNA structures owing to its high conformational plasticity (*Paithankar et al., 2022*). In the current study, we have characterized the interaction of TRBP2-dsRBD2 with the same set of topologically different dsRNAs (wt miR-16-1 and mutants; sequences mentioned in *Supplementary file 1b*) used for TRBP2-dsRBD1 through $^1$H-$^{15}$N HSQC-based NMR titrations shown in *Figure 6—figure supplement 1*. The NMR investigation regarding the interaction of TRBP-dsRBD2 with the wt miR-16-1 and the three mutant dsRNAs revealed only chemical shift perturbations (<0.1 ppm) in the presence of four topologically different RNAs as shown in *Figure 6—figure supplement 1*. This indicated that the tertiary structure/fold of the protein remained unperturbed upon RNA binding. The fact that the structure remains unperturbed is consistent with the previous findings reported by *Yamashita et al., 2011*, where the authors compared the solution structure of TRBP2-dsRBD2 structure with the crystal structure of GC10RNA-bound TRBP2-dsRBD2 (PDB ID: 3ADL) (*Yang et al., 2010*) and concluded that the structure of the dsRBD remains unperturbed upon RNA-binding. However, TRBP2-dsRBD2, in some cases, has shown the presence of chemical shift perturbation <1.0 ppm at the three RNA-binding regions upon binding with RNA (*Benoit et al., 2013*; *Masliah et al., 2018*), indicating subtle conformational

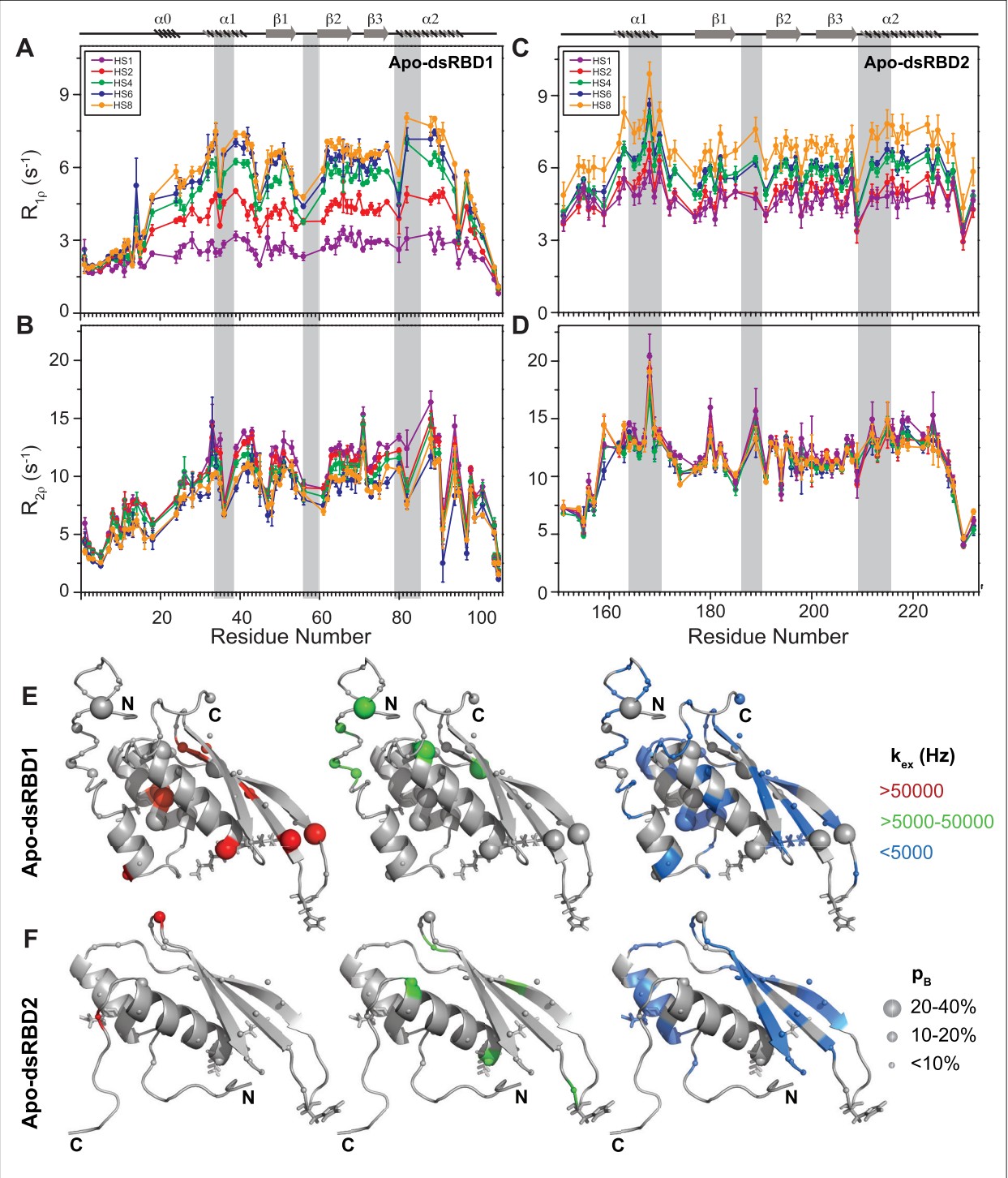

**Figure 4.** Comparison of slow dynamics (μs-ms timescale) for dsRBD1 and dsRDB2 of TRBP2 as measured by NMR relaxation dispersion experiments. (**A**) $R_{1\rho}$ data, (**B**) $R_{2\rho}$ data, recorded on apo $^{15}$N-TRBP2-dsRBD1, (**C**) $R_{1\rho}$ data, (**D**) $R_{2\rho}$ data, recorded on apo $^{15}$N-TRBP2-dsRBD2 using heteronuclear adiabatic relaxation dispersion (HARD) experiments on a 600-MHz NMR spectrometer at 298 K, plotted against residue numbers. An increase in the spin-lock field strength is achieved by an increase in the stretching factor of the adiabatic pulse used to create the spin lock, *n* (in HS*n*). The secondary structure has been depicted on the top, and three RNA-binding regions have been highlighted using vertical grey bars. Mapping of conformational exchange parameters (exchange rate constant between the ground state and excited state ($k_{ex}$), and excited state population ($p_B$)) obtained by fitting the above-described data to a two-state model on the CS-Rosetta structures of (**E**) apo-dsRBD1, and (**F**) apo-dsRBD2. Residues have been marked in different colors to highlight the distribution of $k_{ex}$ values, and the diameters of the sphere indicate the extent of $p_B$ along the protein backbone. The RNA-binding residues have been depicted in stick mode. The analysis was done for 105 residues of TRBP2-dsRBD1 (1–105 aa) and 80 residues of

*Figure 4 continued on next page*

changes at the RNA-binding interface (*Yamashita et al., 2011*; *Yang et al., 2010*). Furthermore, there were two intriguing observations in the $^1$H-$^{15}$N HSQC-based titrations. First, the amide signals were getting broadened at as low as 0.1 RNA equivalents, suggesting the slow-to-intermediate timescale of binding as has also been observed previously by our and other research groups in TRBP2 dsRBD1, Staufen dsRBD3, hDus2 dsRBD, MLE dsRBD2, DBR4 dsRBD1, PKR dsRBD, and Dicer dsRBD (*Ankush Jagtap et al., 2019*; *Bou-Nader et al., 2019*; *Chiliveri et al., 2017*; *Paithankar et al., 2022*; *Ramos et al., 2000*; *Ucci et al., 2007*; *Wostenberg et al., 2012*; *Yadav et al., 2020*). Second, not only the

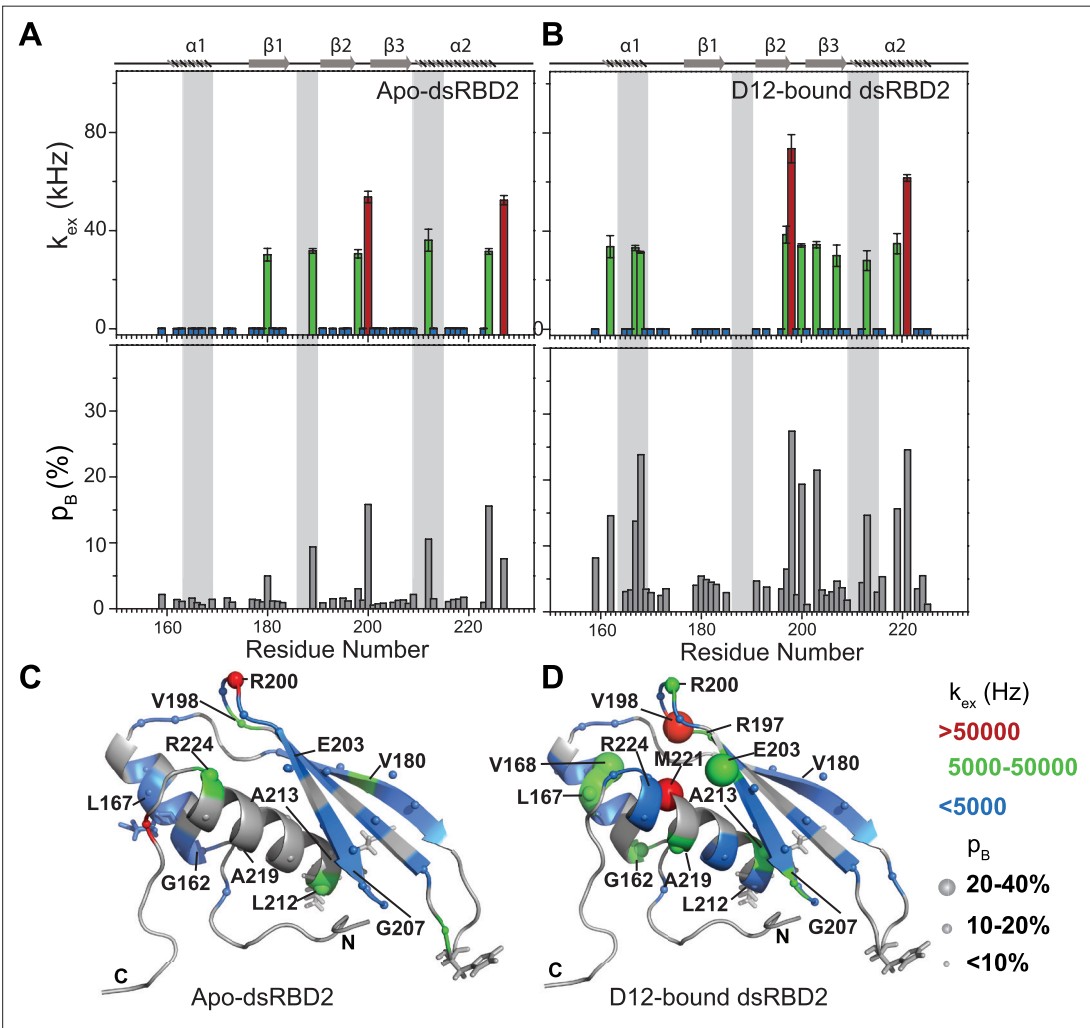

**Figure 5.** Conformational exchange in (**A**) apo- and (**B**) D12-bound TRBP2-dsRBD2. Top panel: The exchange rate constant between the ground state and excited state ($k_{ex}$); bottom panel: excited state population ($p_B$) as obtained by the geometric approximation method, using the heteronuclear adiabatic relaxation dispersion (HARD) experiment, plotted against residue numbers. Mapping of core $k_{ex}$, and $p_B$, on the CS-Rosetta structure of apo TRBP2-dsRBD2, as extracted for (**C**) apo TRBP2-dsRBD2 and (**D**) D12-bound TRBP2-dsRBD2. Different colors highlight the distribution of $k_{ex}$ values, and the sphere's diameter indicates the extent of $p_B$ along the protein backbone. The RNA-binding residues have been depicted in stick mode. The analysis was done for 68 residues of dsRBD2 (159-227 aa) in -apo and RNA bound state. The errors in different fitted parameters were obtained using Monte-Carlo simulations (n=500) and the duplicate relaxation data points (two delays in duplicate).

The online version of this article includes the following figure supplement(s) for figure 5:

**Figure supplement 1.** Heteronuclear adiabatic relaxation dispersion (HARD) experiments recorded on $^{15}$N-TRBP2-dsRBD2 in the presence of D12-RNA on the 600 MHz NMR spectrometer at 298 K.

reported RNA-binding residues but the entire backbone was undergoing line broadening, suggesting the presence of RNA-induced motions in the entire backbone. Since the protein is not saturated with the RNAs at 0.1 RNA equivalents (considering a reported $K_d$ = 1.7 µM for a 22-bp dsRNA (**Acevedo et al., 2015**), [protein] = 50 µM, [RNA] = 5 µM, fraction bound of protein <10%) we can rule out an increase in the size due to the formation of a stable protein–RNA complex causing line broadening. Upon excess addition of RNA (to 1 equivalent of miR-16-1-M), the backbone amide signals were not recovered (**Figure 6—figure supplement 1**, top left panel), indicating the RNA–protein interaction does not seem to come out of the local minima of slow-to-intermediate exchange regime.

The amide NMR signals were, however, partially recovered by shortening the length of the RNA duplex to 12 bp D12 RNA. The line broadening was delayed till 0.35 equivalents of D12 RNA as compared to 0.1 equivalents for the longer RNAs (**Figure 6—figure supplement 2**). The excessive line broadening all along the backbone and recovery of the same by reducing the length of the RNA

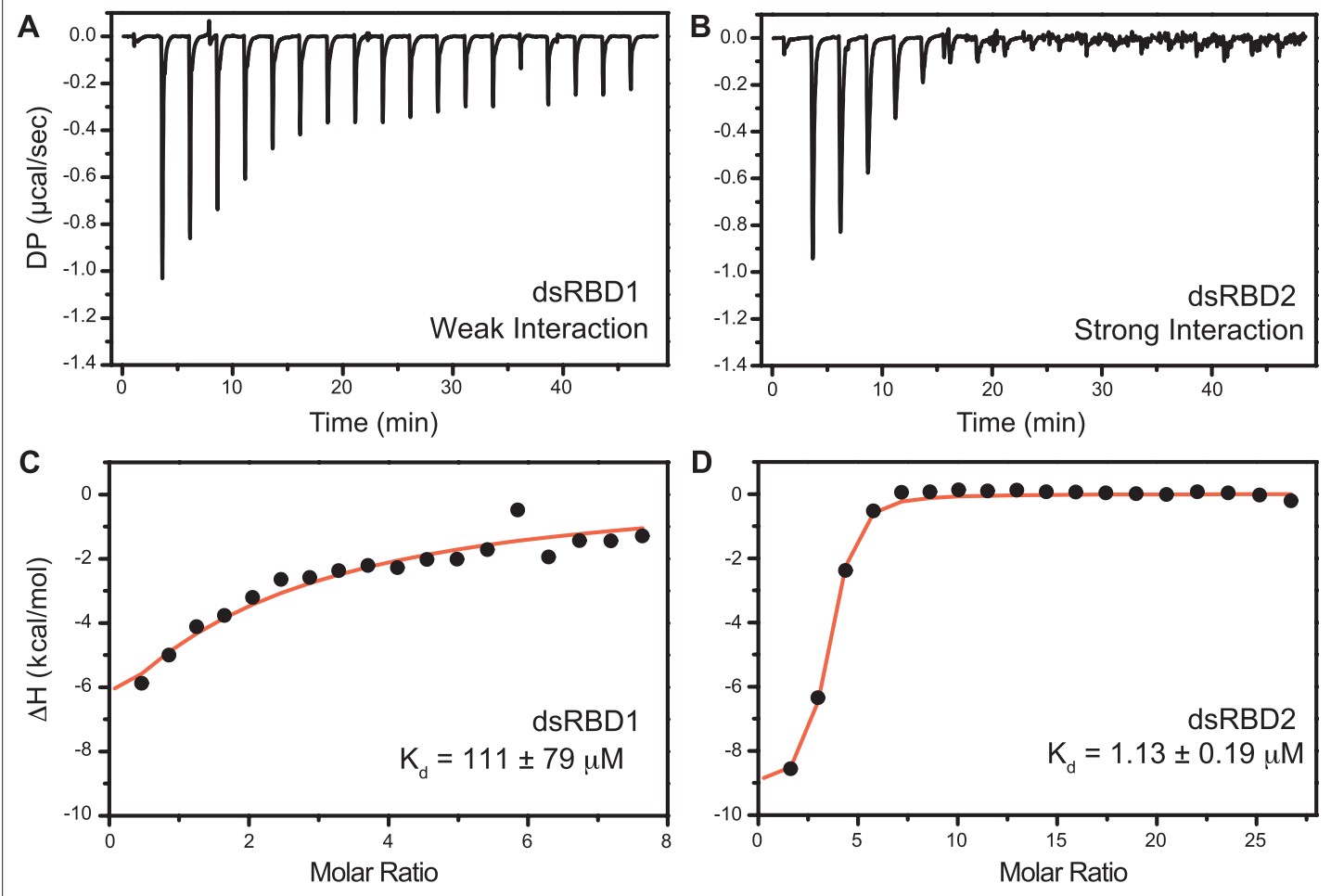

**Figure 6.** Isothermal titration calorimetry (ITC)-based binding study of D12 duplex RNA with TRBP2-dsRBD1 and TRBP2-dsRBD2. Top panel: the raw differential potential for each injection is plotted against the titration time for (**A**) TRBP2-dsRBD1 and (**B**) TRBP2-dsRBD2. Bottom panel: the integrated heat (enthalpy change) upon each injection (black dots) and the data fit for a single set of binding sites (red line) plotted against per mole of injectant for (**C**) TRBP2-dsRBD1 and (**D**) TRBP2-dsRBD2.

The online version of this article includes the following figure supplement(s) for figure 6:

**Figure supplement 1.** Titration of [15]N-TRBP2-dsRBD2 with miR-16-1-M (an overlay of [1]H-[15]N -HSQC spectra of apo-protein (green) RNA:protein (R:P) = 0.25:1 (red) and (R:P) = 1:1 (black)); miR-16-1-B, miR-16-1-D, and miR-16-1-A (wt) [overlaid [1]H-[15]N -HSQC spectra of apo-protein (green), (R:P) = 0.1:1 (red) and (R:P) = 0.2:1 (black)].

**Figure supplement 2.** Titration of [15]N-TRBP2-dsRBD2 with D12 RNA.

**Figure supplement 3.** Normalized NMR peak intensity from [1]H-[15]N HSQC spectra of TRBP2-dsRBD2 plotted against protein: D12 RNA ratio from the titration carried out in **Figure 6—figure supplement 2**.

hints toward the presence of the phenomenon of protein diffusing over the length of the RNA, as has been reported by Koh *et. al.* earlier using smFRET (*Koh et al., 2013*). Additionally, the phenomenon of diffusion along the length of RNA has been hinted at in other dsRBPs like Loqs-PD dsRBD (*Tants et al., 2017*), MLE dsRBD (*Ankush Jagtap et al., 2019*), DRB4 dsRBD1 (*Chiliveri et al., 2017*), and RDE4 (*Chiliveri and Deshmukh, 2014*). To extract the equilibrium dissociation constant ($K_d$) for the D12–TRBP2–dsRBD2 interaction, residue-wise peak intensities were plotted against the RNA concentration and tried to fit to the binding isotherm for one-site binding (*Figure 6—figure supplement 3*). Due to extensive line broadening, there was a lack of data at the inflection point, affecting the data fitting. Hence, the fitted parameters had large errors and remained inconclusive and thus are not reported here.

The isothermal titration calorimetry (ITC)-based study was performed with D12 duplex RNA (*Figure 6*) to get the equilibrium dissociation constant ($K_d$) of the two N-terminal dsRBDs, change in enthalpy upon RNA binding ($\Delta H$), and stoichiometry ($n$) (*Supplementary file 1j*). The differential potential against the time plot indicates that RNA did not get saturated with TRBP2-dsRBD1 (*Figure 6A*), while it got saturated with TRBP2-dsRBD2 as early as at the eighth injection (*Figure 6B*). The TRBP2-dsRBD1 data could not be fit conclusively to any binding model ($K_d$ has large errors in these data) as depicted in *Figure 6C*. Our studies showed that TRBP2-dsRBD2 binds to D12 RNA with an average $K_d$ of 1.18 ± 0.32 µM (*Figure 6D* and *Supplementary file 1j*), while TRBP2-dsRBD1 did not show any significant heat exchange (weak binding) during the titration. The integrated change in injection enthalpy ($\Delta H$) versus the molar ratio of the reactants yielded an average $\Delta H$ of −10.12 ± 0.43 kcal/mol, suggesting an enthalpy-driven binding event. The entropy $\Delta S$ of the binding event was negative, indicating the absence of hydrophobic interaction between the D10 RNA and dsRBD2. ITC study was repeated for TRBP2-dsRBD1 with as high as 22 folds excess of protein, but RNA remained unsaturated. These results strongly suggest that TRBP2-dsRBD2 binds more efficiently to a small perfect A-form duplex, D12 RNA, than TRBP2-dsRBD1.

## Dynamics perturbations of TRBP2-dsRBD2 at various timescales in the presence of RNA ligand

Minute structural changes in TRBP2-dsRBD2 in the presence of RNA necessitated a deeper insight into the perturbations of conformational dynamics at multiple timescales. Backbone $^1$H-$^{15}$N dynamics measurements on TRBP2-dsRBD2 were carried out in the limiting concentration (RNA:protein = 0.05) of short duplex D12RNA as an increase in the RNA length and an increase in the RNA concentration, both caused line-broadening impacting dynamics measurements. In the presence of D12 RNA, the average $R_1$ rates (apo = 1.43 ± 0.05 s$^{-1}$; bound = 1.37 ± 0.07 s$^{-1}$) and nOe values (apo = 0.73 ± 0.03; bound = 0.72 ± 0.076) remained unperturbed, while there was a significant increase in the average $R_2$ rates (apo = 10.92 ± 0.37 s$^{-1}$; bound = 20.04 ± 0.11 s$^{-1}$) (*Figure 3* and *Supplementary file 1k*). The apparent increase in the $R_2$ rates (=$R_2$*+ $R_{ex}$) hinted toward a perturbation in the µs–ms timescale dynamics, to which only $R_2$ rates are sensitive. This phenomenon could be attributed to either an increase in intrinsic $R_2$* resulting from an increase in the residence time of the RNA on the protein, or an increase in the $R_{ex}$ component caused by a chemical exchange between apo- and D12-bound state, or RNA-induced conformational exchange in the protein. Interestingly, such a perturbation in $R_2$ rates was not found for the dsRBD1 upon RNA binding (*Paithankar et al., 2022*). The core N- and C-terminal residues (S151, Q154, Q155, S156, E157, N159, V225, V228, L230, and A232), and a few residues in the vicinity of loop regions (E177, E183, G185, E191, and S209) harbored a lower than average bound-state $R_2$ rates than the rest of the core indicating a faster motion induced in these residues (*Figure 3*). On adding RNA, the $R_2$ rates and nOe values of the N- and C-terminal residues (S156, E157, T227, V228, L230, and A232), β1 (E177), the loop 2 region (E183, G185, 189, and E191), and the loop 4 region (S209) decreased, indicating further enhanced flexibility in one of the vital RNA-binding region (*Figure 3* and *Supplementary file 1k*).

The extended model-free analysis of apo and RNA-bound TRBP2-dsRBD2 suggested that the anisotropic (ellipsoid) diffusion model was the best fit for the global motion of them both. The global rotational correlation time ($\tau_c$) of core TRBP2-dsRBD2 in the presence of RNA was 10.9 ns as against 7.3 ns for the apo-dsRBD2 core, indicating an apparent increase in molecular weight. The overall higher $S^2$ values for TRBP2-dsRBD2 indicated a rigidification of the backbone amide vectors in the presence of D12 RNA (*Figure 3—figure supplement 4* and *Supplementary file 1l*). A few residues

in the $\alpha_1$ (E166) and $\alpha_2$ (L212, N216, and L223) regions exhibited a $R_{ex}$ component >1 s$^{-1}$, lying in the vicinity of the reported RNA-binding regions 1 and 3 (*Supplementary file 1m*). The number of residues with a significant (>2 s$^{-1}$) $R_{ex}$ – implying the presence of µs–ms timescale motions – increased in the case of bound state. $R_{ex}$ was found to be induced in the $\alpha_1$ (V168), $\beta_1$ (Y178), $\beta_2$ (T195), $\beta_3$ (I204), and $\alpha_2$ (N216 and L223) region, thereby indicating that not only the RNA-binding residues but the rest of the core might play a role while interacting with dsRNA substrates (*Supplementary file 1m*). The presence of $R_{ex}$ at multiple sites (in addition to RNA-binding regions) rules out the chemical exchange between apo- and D12-bound state contributing to $R_{ex}$, thereby implying a presence of RNA-induced conformational exchange is the predominant contributor to $R_{ex}$ and to increased line broadening (or apparent $R_2$ rates, discussed in the NMR-based titration paragraph).

The effective transverse relaxation rates, $R_{2,eff}$, for TRBP2-dsRBD2, were plotted against the CPMG frequencies ($\nu_{CPMG}$) in the presence of D12 RNA (*Figure 3—figure supplement 5* and *Supplementary file 1n*). None of the residues in either condition showed effective dispersion in the $R_{2,eff}$ rates with the increase in $\nu_{CPMG}$, suggesting motions in the timescale sensitive to this experiment remain absent in D12RNA-bound TRBP2-dsRBD2.

Most importantly, HARD NMR experiments recorded for TRBP2-dsRBD2 in the presence of D12-RNA showed fascinating results (*Figure 5—figure supplement 1* and *Supplementary file 1o and 1p*). In the presence of RNA, there was massive induction of 10 µs to 10 ms timescale conformational dynamics as reflected by the increase in the number of residues having significant $k_{ex}$ (*Supplementary file 1q*). The extent of enhancement in $k_{ex}$ varied along the backbone of the core protein. For instance, a $k_{ex}$ of 5000–50,000 kHz with 10–20% $p_B$ (green spheres) was observed in $\alpha_1$ (G162 and L167), loop3 (R200), and $\alpha_2$ (A213 and A219) and 20–40% $p_B$ in $\alpha_1$ (V168), and $\beta_3$ (E203), depicted by the medium and big green spheres, respectively (*Figure 5B, D* and *Supplementary file 1q*). The residues V198 in loop 3 and M221 in the $\alpha_2$ region exhibited the presence of the highest frequency of motion with a $k_{ex}$ > 50,000 Hz. Intriguingly, among these residues, L167, V168 ($\alpha_1$), and A213 ($\alpha_2$) lie in the reported RNA-binding regions 1 and 3, respectively. L167 and V168 are adjacent to the key RNA-binding residue E165 (region 1), which interacts directly with the RNA minor groove. A213 precedes the important KR-helix motif in the $\alpha_2$ region of dsRBD2. The K214 and R215 residues make ionic interactions with the negatively charged phosphate backbone of the RNA major groove. The interaction between the RNA and the RNA-binding residues of the protein might induce conformational exchange within the nearby residues like in the case of G162 (region 1), A219, and M221 (region 2). Additionally, residues

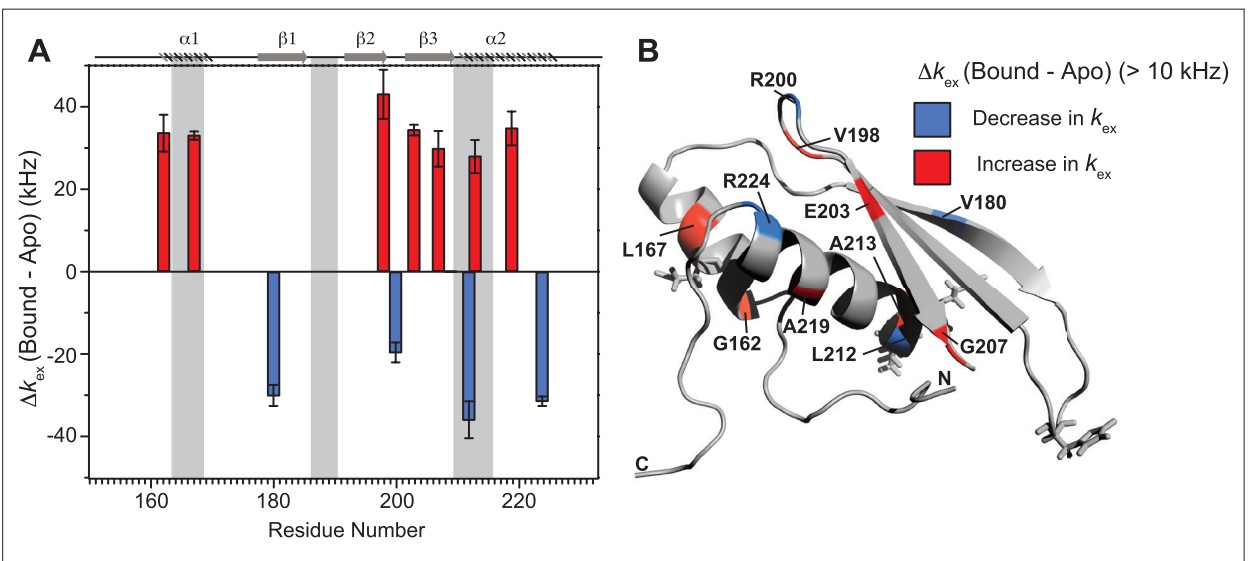

**Figure 7.** Conformational exchange perturbations in core TRBP2–dsRBD2 in the presence of D12 RNA. (**A**) $\Delta k_{ex}$ (D12-bound – apo) TRBP2–dsRBD2 plotted against residue numbers. The secondary structure has been shown on the top, and three RNA-binding regions have been highlighted using vertical grey bars. Only residues having significant perturbation ($\Delta k_{ex}$ > 10 kHz) have been plotted, where an increase is shown in red, and a decrease is shown in blue, (**B**) An increase in $k_{ex}$ (red) and a decrease (blue) in the presence of D12 RNA indicated on the backbone of the CS-Rosetta structure of apo TRBP2–dsRBD2. The RNA-binding residues have been depicted in stick mode in the tertiary structure.

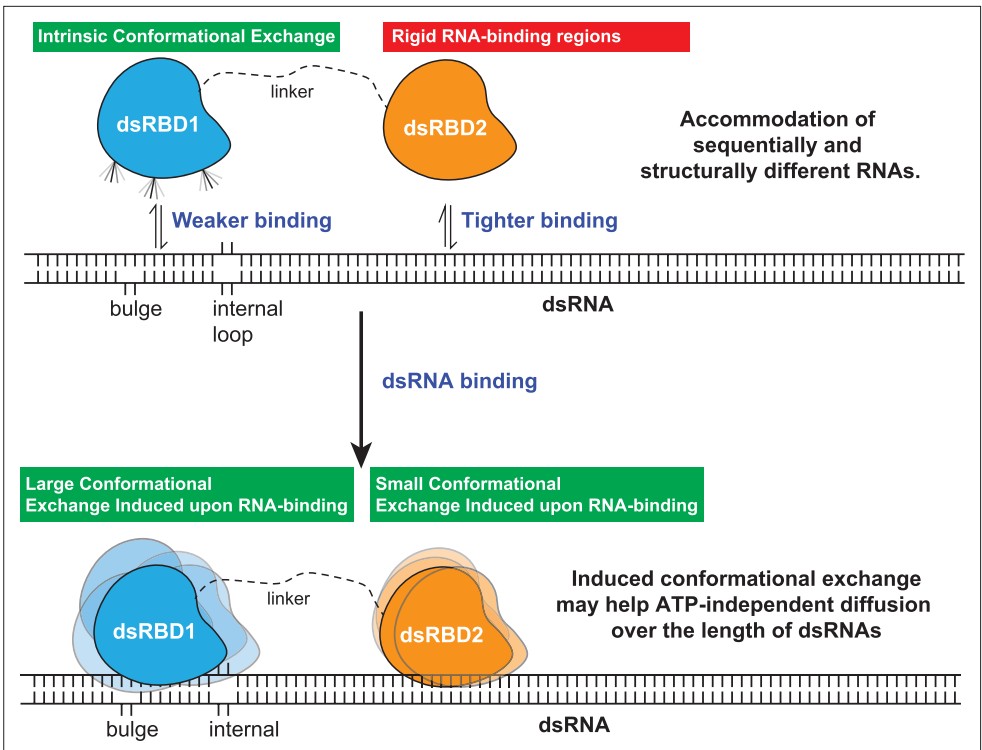

**Figure 8.** The model proposed for the two type-A dsRBDs in TRBP2 protein. dsRBD2 with rigid and conserved RNA-binding regions is able to bind the RNA tightly, whereas dsRBD1 with high conformational exchange is able to recognize different RNA structures (often with bulges and internal loops). Following this, the two dsRBDs upon contacting the RNA undergoes enhanced conformational exchange at different extents. This enhanced conformational exchange coupled with differential binding affinity toward dsRNA might enable the tandem dsRBDs to move along the backbone of the RNA molecule, leading to ATP-independent diffusion.

like R197 ($\beta_2$); V198 and R200 (L3); E203 and G207 ($\beta_3$) are further from RNA-binding regions with $k_{ex} >$ 5000 Hz. Thus, the ligand affected not only the RNA-binding residues but the entire protein.

Notable perturbations in terms of $\Delta k_{ex}$ ($k_{bound} - k_{apo}$) > 10,000 Hz were mapped on the structure of the protein (**Figure 7**). There was an enhancement of $k_{ex}$ in the residues lying in $\alpha_1$ (G162 and L167), loop 3 (V198), $\beta_3$ (E203 and G207), and $\alpha_2$ (A213 and A219) region. Concomitantly, suppression of exchange was observed in $\beta_1$ (V180), loop 3 (R200), and $\alpha_2$ (L212 and R224) regions. A tantalizing relay of exchange was seen between two residue pairs in close spatial proximity. For instance, exchange was induced in V198 and quenched in R200 in the loop 3 region. Similarly, in the case of A213 and L212 falling in the N-terminal RNA-binding region 3, and A219 and R224 lying in the C-terminal of $\alpha_2$, the former of the pairs underwent an increase in $k_{ex}$ while the latter observed a decrease. Thus, the enhancement of RNA-induced conformational exchange was accommodated by allosteric quenching of the same. Altogether, there was a significant induction of 10 µs to 10 ms timescale motions over the entire backbone of TRBP2-dsRBD2 in the presence of RNA. This induction of motions could be ascribed to either an exchange between the apo- and RNA-bound state of the protein, conformational dynamics in the RNA-bound state of the protein, or the previously reported diffusion motion of the protein on the dsRNA (**Koh et al., 2013**). Interestingly, in TRBP2-dsRBD2, the extent of dynamics perturbation, in terms of $\Delta k_{ex}$, was significantly lower (10–50 kHz) than TRBP2-dsRBD1 where at least 10 residues underwent a $\Delta k_{ex} > 50$ kHz in the presence of small dsRNA. Moreover, a relay of exchange was evident only in loop 3 and $\alpha_2$ regions of dsRBD2 and not in $\alpha_1$ region, which was observed earlier in dsRBD1. Summing up, in the presence of RNA, dsRBD2 undergoes enhanced conformational exchange in the 10 µs–10 ms timescale. When compared to dsRBD1, dsRBD2 samples limited conformational space both in the absence and presence of D12-RNA.

Thus, our study has led us to propose a model involving two tandem type-A dsRBDs in TRBP2. According to our model, these dsRBDs work synergistically to recognize, bind, and assist associated

proteins (like Dicer) in processing the incoming dsRNA ligand (*Figure 8*). The first dsRBD, TRBP2-dsRBD1, has remarkable flexibility, which allows it to recognize a wide range of structurally and sequentially diverse dsRNAs. On the other hand, TRBP2-dsRBD2 firmly adheres to the RNA ligand due to its conserved RNA-binding stretches and relatively high rigidity. During their interaction with the RNA, both dsRBDs experience enhanced conformational changes on a fast timescale (picoseconds to nanoseconds) and a moderate timescale (microseconds to milliseconds), as observed through nuclear spin relaxation and rotating frame relaxation dispersion measurements. These dynamic changes likely enable the dsRBDs to diffuse along the length of the RNA, which explains the broadening of the signals observed during titration with longer RNAs. This diffusion movement is likely crucial for other associated proteins, such as Dicer (*Fareh et al., 2016*; *Lee and Doudna, 2012*; *Wilson et al., 2015*), to carry out precise microRNA biogenesis. By understanding these processes, we can gain insights into the intricate mechanisms involved in RNA processing, which has implications for different biological processes.

## Conclusions

It has been long established that despite engaging with a wide array of dsRNA molecules exhibiting diverse structures and sequences, dsRBDs remain unaffected in their own structural conformation. However, the more significant question that has persisted is whether protein dynamics itself plays a crucial role in these intriguing interactions. Though variability in the binding affinity for the two type-A dsRBDs of TRBP has been reported, the specific contribution of conformational dynamics has remained unexplored until now. In the present investigation, we have exclusively delved into the role of conformational dynamics of the two type-A dsRBDs of TRBP2 in RNA recognition and binding. Our findings have unveiled that TRBP2-dsRBD2 samples a limiting conformational space in solution, and it is dsRBD1 that recognizes sequentially and structurally diverse RNA substrates through its high conformational plasticity. While dsRBD1 explores and engages dsRNA via its dynamically interactive RNA-binding surface, dsRBD2 holds the RNA in position via stronger canonical contacts. Once bound to the RNA, the ensuing conformational exchange in both dsRBD1 and dsRBD2 might facilitate the domains to diffuse over the length of the RNA freely, thus playing a pivotal role in assisting Dicer-mediated differential cleavage of RNA. Thus, this study not only adds valuable insights into the mechanics of RNA–protein interactions but also underscores the significance of conformational dynamics in dictating the functional outcome in such intricate biological processes.

Exploring the intricacies of RNA–protein interactions by delving into dynamics-based measurements across diverse RNA shapes within small RNA duplexes could offer invaluable insights into the differential recognition and binding by dsRBD1 and dsRBD2. The experimental challenge lies in the meticulous design of RNA sequences that maintain duplex stability amid the presence of bulges and internal loops of varying lengths and sequences. The task is further complicated by managing RNA length judiciously, as longer sequences can introduce line broadening in NMR spectroscopy. Navigating these complexities in the future promises to unravel the subtleties of the RNA–protein interactions at a molecular level.

# Materials and methods

**Key resources table**

| Reagent type (species) or resource | Designation | Source or reference | Identifiers | Additional information |
|---|---|---|---|---|
| Gene (*Homo sapiens*) | TARBP2 | GenBank | HGNC:HGNC :11569 | Gene ID: 6895 (Uniprot ID: Q15633) |

*Continued on next page*

*Continued*

| Reagent type (species) or resource | Designation | Source or reference | Identifiers | Additional information |
|---|---|---|---|---|
| Strain, strain background (*Escherichia coli*) | BL21(DE3) and DH5a | Others | | Chemically competent cells obtained from Dr. Gayathri Pananghat (Indian Institute of Science Education and Research Pune, Pune, India) as kind gift and further amplified in our lab. |
| Recombinant DNA reagent | pHMGWA (plasmid) | Others | | TRBP2-dsRBD2 (154–234 aa) was cloned in pHMGWA vector via Gateway cloning and was obtained as a kind gift from N. L. Prof. Jennifer Doudna (University of California, Berkeley, CA, USA) |
| Sequence-based reagent | miR-16-1-A; miR-16-1-D; miR-16-1-M; miR-16-1-B; D12 RNA | Integrated DNA Technologies (Coralville, IA, USA) or GenScript Biotech Corporation (Piscataway, NJ, USA) | RNA oligos | Guide:UAGCAGC ACGUAAA UAUUGGCG Passenger:CCAGU AUUAACUG UGCUGCUGAA; Guide:UAGCAGC ACGUAAA UAUUGGCG Passenger: CCAGUAU UUACGUGCUG CUGAA; Guide:UAGCAGCA CGUAAAUAUUG GCG Passenger: CCAGUAUUAACG UGCUGCUGAA; Guide:UAGCA GCACGUAAAUA UUGGCG Passenger: CCAGUAU UAACGUGCU GCUGAA; Guide:CGUAAA UAUUCG Passenger:CGAGUA UUUACG |
| Chemical compound, drug | $^{15}NH_4Cl$; $^{13}C$-glucose | Cambridge Isotope Laboratories | NLM-467-50; CLM1396-10 | |
| Software, algorithm | MicroCal PEAQ-ITC analysis software | Malvern Panalytical, Malvern, UK | | |
| Software, algorithm | NMRPipe | https://doi.org/10.1007/BF00197809 | | |
| Software, algorithm | SPARKY (version 3.115) | https://doi.org/10.1093/bioinformatics/btu830 | | |
| Software, algorithm | CARA | ISBN 3-85600-112-3 | | |
| Software, algorithm | Relax v4.0.3 software | https://doi.org/10.1093/bioinformatics/btu166 | | |

## Protein overexpression and purification

The TRBP2-dsRBD1 (1–105 aa) and TRBP2-dsRBD2 (154–234 aa) cDNA clones were obtained as a kind gift from N. L. Prof. Jennifer Doudna (University of California, Berkeley, CA, USA). The residue numbers in the two constructs (*Figure 1A*) have been mentioned in reference to the full-length TRBP2

sequence (1–366 aa, Uniprot ID: Q15633-1). The cDNA for TRBP2-dsRBD2 was cloned in pHMGWA vector (Amp$^R$), and was expressed as a fusion protein having N-terminal His$_6$-Maltose-binding protein (MBP) tag-TEV protease cleavage site followed by the protein of interest. Treatment with TEV protease during purification resulted in non-native Ser–Asn–Ala residues at the N-terminal to TRBP2-dsRBD2 (154–234), which were excluded from all the NMR-based dynamics studies. For the NMR experiments, $^{15}$N- and $^{15}$N-$^{13}$C-labeled (as required) TRBP2-dsRBD2 protein was prepared using $^{15}$NH$_4$Cl and $^{13}$C-glucose (Cambridge Isotope Laboratories) as a sole source of nitrogen and carbon, respectively, in M9 minimal medium.

TRBP2-dsRBD1 was overexpressed and purified as previously described (*Paithankar et al., 2018*). Briefly, the pHMGWA plasmid containing the cDNA for the TRBP2-dsRBD2 was transformed into *E. coli* BL21 (DE3) cells, and plated on an LB agar plate (containing 100 µg/ml ampicillin) and incubated at 37°C for 12 hr. A single isolated colony was inoculated into 20 ml LB broth (containing 100 µg/ml ampicillin) and incubated for 12 hr at 37°C, 225 rpm to initiate a starter culture that was eventually used to inoculate 2 l LB broth (containing 100 µg/ml ampicillin) and incubated at 37°C, 225 rpm till the OD$_{600}$ reached to 0.8–1.0. For induction, IPTG (isopropyl β-d-1-thiogalactopyranoside) was added at the final concentration of 1 mM, and the culture was further incubated at 37°C, 225 rpm for another 8 hr. The cells were harvested by centrifuging the culture at 4500 × *g*, 4°C for 20 min and resuspended in 25 ml of buffer A (20 mM Tris–Cl, pH 7.5, 500 mM NaCl, 10% glycerol, 5 mM DTT :dithiothreitol, 10 mM imidazole). To the resuspended cells, lysozyme (Sigma-Aldrich) was added to a final concentration of 50 µg/ml and incubated in ice for 30 min. Post-incubation, 1% Triton X-100, 100 µl of 1 mM PMSF (phenylmethylsulfonyl fluoride), and 10× Protease Inhibitor Cocktail (PIC, Roche) (2 ml per 4 g of cell pellet) were added to the cell suspension. The partially lysed cells were sonicated using an ultrasonic sonicator microprobe at 60% amplitude, with 5 s ON and 10 s OFF pulse, for a period of 60 min in an ice bath for complete lysis. The cell lysate was centrifuged at 15,000 × *g* for 2 hr, at 4°C, to obtain the total soluble protein (TSP). The TSP was then circulated through a pre-equilibrated (buffer A) Ni-NTA column (HisTrap, 5 ml, GE HealthCare) for 4 hr at 4°C. After equilibrating with the TSP, the column was washed with 40 column volumes (CVs) of buffer A containing 30 mM imidazole to remove the impurities. The fusion protein was eluted with the elution buffer B (buffer A containing 300 mM imidazole). Nucleic acids were removed from the eluted protein using polyethyleneimine precipitation. To remove imidazole, the protein solution was dialyzed against cleavage buffer C (20 mM Tris–Cl, pH 7.5, 25 mM NaCl, 10% glycerol, 5 mM DTT). His$_6$-MBP-tag was cleaved using TEV protease at a final concentration of 1:100 (protease: protein) at 4°C for 18 hr with intermittent mixing. The completion of the cleavage reaction was tested and confirmed by sodium dodecyl sulfate–polyacrylamide gel electrophoresis analysis. The cleavage mix was then passed through a sulphopropyl sepharose column (HiPrep SP FF 16/10 20 ml, GE HealthCare) and washed with buffer C. TRBP2-dsRBD2 protein was eluted from the cation-exchange column using a salt gradient ranging from 25 mM to 1 M NaCl in the buffer C. To further purify, the protein was subjected to size-exclusion chromatography using sephacryl S-100 HR 16/60 column (GE HealthCare). The final purified protein was concentrated to 1 mM using Amicon (3 kDa cutoff, Merck) and exchanged with NMR buffer D (10 mM sodium phosphate, pH 6.4, 100 mM NaCl, 1 mM EDTA:ethylenediaminetetraacetic acid, 5 mM DTT) before recording any experiment.

### Design and preparation of RNA

RNA duplexes miR-16-1-A, miR-16-1-M, miR-16-1-B, and miR-16-1-D were designed and procured as mentioned previously (*Paithankar et al., 2022*). The shorter 12 bp RNA duplex RNA oligo (D12) was designed from miR-16-1-D and was procured from either Integrated DNA Technologies (Coralville, IA, USA) or GenScript Biotech Corporation (Piscataway, NJ, USA). All the RNA sequences used in this study have been listed in *Supplementary file 1b*.

The preparation of RNA duplexes and the confirmation of their formation was performed, as described previously (*Paithankar et al., 2022*). Briefly, the respective guide and passenger RNA listed in *Supplementary file 1b* were mixed together in a 1:1 ratio, followed by denaturation at 90°C for 10 min and then cooling at 4°C for 30 min. The annealing of all four RNAs (miR-16-1-A, miR-16-1-M, miR-16-1-B, and miR-16-1-D) (*Paithankar et al., 2022*) and D12 RNA was confirmed by $^1$H-NMR by observing the imino proton signals (*Figure 1—figure supplement 3*). The annealed samples were maintained in buffer D for all data measurements.

## Size-exclusion chromatography – multiple angle light scattering

SEC-MALS experiments were performed using an S75 column (Superdex 75 10/300 GL 24 ml, GE HealthCare), Agilent HPLC system (Wyatt Dawn HELIOs II) and a refractive index detector (Wyatt Optilab T-rEX). The system was first calibrated by injecting 100 µl of 30 µM bovine serum albumin solution (Thermo Scientific). Post calibration, 100 µl protein samples were injected (in duplicate) at a concentration of 0.8 mM for TRBP2-dsRBD2. The respective molar mass values of the peaks were calculated using the Zimm model in ASTRA software version 7 (Wyatt Technologies).

## Isothermal titration calorimetric binding assays

All ITC experiments were performed using a MicroCal PEAQ-ITC calorimeter (Malvern Panalytical, Malvern, UK) operating at 25°C. The final RNAs and protein solutions used for the assays were prepared in buffer D. The D12 dsRNA was used at a concentration of 10 or 20 µM in the sample cell. TRBP2-dsRBD1 concentration was varied from 5 to 19 folds of RNA, whereas, in the case of TRBP2-dsRBD2, it was varied from 10 to 18 folds. The first injection was 0.4 µl (discarded for data analysis), which was followed by eighteen 2 µl injections. All the ITC data were measured in triplicate.

Data were fitted with a single-site-binding model using the MicroCal PEAQ-ITC analysis software (Malvern Panalytical, Malvern, UK) to extract the equilibrium dissociation constant ($K_d$), stoichiometry ($n$), and change in enthalpy ($\Delta H$). The final values of the thermodynamic parameters are given as the average of triplicate measurements (*Supplementary file 1j*).

## NMR spectroscopy

All the NMR experiments were recorded at 298 K either on: (1) Ascend Bruker AVANCE III HD 14.1 Tesla (600 MHz) NMR spectrometer equipped with a quad-channel ($^1$H/$^{13}$C/$^{15}$N/$^{19}$F) Cryoprobe (in-house); or (2) Ascend Bruker Avance AV 18.89 Tesla (800 MHz) NMR spectrometer equipped with a triple-channel ($^1$H/$^{13}$C/$^{15}$N) Cryoprobe and a Broad Band Inverse probe (located at National Facility for High-Field NMR at TIFR, Mumbai). The $^1$H-$^{15}$N HSQC spectrum was collected with 2048 and 128 points and 12 and 28 ppm spectral width in $^1$H and $^{15}$N dimensions, respectively, giving an acquisition time of 100 ms in the direct dimension. An inter-scan delay of 1.0 s and 4 scans were used on a 1-mM $^{15}$N-labeled TRBP2-dsRBD2 sample in a 5-mm Shigemi tube (Shigemi Co, LTD, Tokyo, Japan) (*Takeda et al., 2011*). $^1$H-$^{15}$N TOCSY-HSQC (mixing times = 60, 80, and 120 ms) and $^1$H-$^{15}$N NOESY-HSQC (mixing times = 150, 300, and 400 ms) were recorded on a 600-MHz NMR spectrometer on $^{15}$N-labeled TRBP2-dsRBD2 for the backbone assignment of TRBP2-dsRBD2. Furthermore, triple resonance experiments like HNCO, HNCACO, HNCA, HNCOCA, HNN, CBCANH, and CBCA(CO)NH were carried out to make the sequential connections using 1.2 mM of $^{15}$N-$^{13}$C-labeled TRBP2-dsRBD2 sample in a 5-mm Shigemi tube. All the NMR data were processed via TopSpin/NMRPipe (*Delaglio et al., 1995*) and were analyzed in SPARKY/CARA (*Keller, 2004*; *Lee et al., 2015*).

For NMR-based titration assays, $^1$H-$^{15}$N-HSQCs were measured on TRBP2-dsRBD2 (50 µM) with increasing concentrations of duplex RNAs from 0.05 to 0.2 equivalents (in the case of D12 RNA 0.05–5 equivalents) of the protein. After every RNA addition, the protein was allowed to equilibrate for 30 min before acquiring $^1$H-$^{15}$N HSQC. Peak intensities were plotted against the RNA:protein concentration for each residue and were fit to the one-site binding isotherm using the below equation (*Williamson, 2013*):

$$\Delta\delta_{obs} = \Delta\delta_{max}([P]_t + [L]_t + K_d) - [([P]_t + [L]_t + K_d)^2 - 4[P]_t[L]_t]^{1/2}/2[P]_t$$

where $\Delta\delta_{obs}$ is the change in the observed peak intensity from the apo state, $\Delta\delta_{max}$ is the maximum peak intensity change on saturation with ligand, $[P]_t$ and $[L]_t$ are the total concentration of protein and ligand, respectively, and $K_d$ is the equilibrium dissociation constant.

Apo-TRBP2-dsRBD2 nuclear spin relaxation experiments ($R_1$, $R_2$, and [$^1$H]-$^{15}$N nOe) were recorded at two different field strengths (600 and 800 MHz NMR spectrometers) on a 1 mM $^{15}$NTRBP2-dsRBD2 sample in a 5 mm Shigemi tube. $^{15}$N longitudinal relaxation rates ($R_1$) were recorded with 8 inversion recovery delays of 10, 30*, 50, 100, 200, 300, 450*, and 600 ms. $^{15}$N transverse relaxation rates ($R_2$) were recorded with 8 CPMG (Carr-Purcell-Meiboom-Gill) delays of 17, 34*, 51, 68, 85, 102, 136*, and 170 ms. Steady-state [$^1$H]-$^{15}$N heteronuclear nOe experiments were recorded with and without $^1$H saturation with a relaxation delay of 5 s.

All D12-bound TRBP2-dsRBD2 nuclear spin relaxation experiments were recorded in a similar fashion as the apo-protein on a 1-mM $^{15}$N-TRBP2-dsRBD2 in the presence of 50 µM D12 RNA in a 3-mm NMR tube. The $^{15}$N-$R_1$ rates were measured with 5 inversion recovery delays of 10, 30*, 70, 150, and 600 ms on 600 MHz spectrometer and 10, 70*, 150, 300, and 600* ms on 800 MHz spectrometer. The $^{15}$N-$R_2$ rates were measured with five CPMG delays of 17, 51, 85, 136*, and 170 ms on 600 MHz spectrometer and with four CPMG delays of 17, 34*, 51, and 68 ms on 800 MHz spectrometer with a CPMG loop length of 17 ms.

For $^{15}$N relaxation dispersion measurement, a constant time CPMG experiment (*Tollinger et al., 2001*) was recorded on a 600-MHz NMR spectrometer. CPMG relaxation dispersion experiments for apo and D12-bound $^{15}$N-TRBP2-dsRBD2 were acquired separately, consisting of 15 data points with $\nu_{CPMG}$ values of 25, 50, 75, 125*, 175, 275, 375*, 525, 675, 825*, and 1000 Hz at a constant relaxation time – $T_{relax}$ (40 ms).

HARD experiments (*Mangia et al., 2010*; *Traaseth et al., 2012*) were recorded on apo and D12-bound $^{15}$N-TRBP2-dsRBD2 on the 600 MHz NMR spectrometer, as described previously (*Paithankar et al., 2022*). The relaxation delays used for $R_{1\rho}$ were 0, 16, 32, 64, 96, and 128 ms for apo-; and 0, 16, 32, 64, and 112 ms for D12-bound $^{15}$N TRBP2-dsRBD2. For $R_{2\rho}$ experiments, the relaxation delays used were 0, 16, 32, and 64 ms for apo and 0, 16, and 32 ms for D12-bound $^{15}$N-TRBP2-dsRBD2. $R_1$ experiments were acquired similarly to $R_{1\rho}$ and $R_{2\rho}$ experiments without using the adiabatic pulse during evolution. The delays used for the $R_1$ experiment were 16, 48, 96, 160, 224, 320, 480, and 640 ms for both apo- (*Supplementary file 1r*) and RNA-bound protein (*Supplementary file 1s*) samples.

All relaxation experiments were measured using a single scan interleaving method, and the order of delays was randomized. The time periods marked with an asterisk have been recorded in duplicates for error estimation. An inter-scan delay of 2.5 s was used for all the above-mentioned relaxation experiments (unless mentioned otherwise).

## NMR relaxation data analysis

NMR spectra were processed with either TopSpin 3.5pl6 or NMRPipe and visualized through SPARKY (version 3.115). For the backbone resonance assignment, CARA (http://cara.nmr.ch/doku.php) was used. All the input NMR spectra were fed in the form of either ucsf or 3rrr file formats. Manual peak picking was done for double and triple-resonance NMR experiments using a synchroscope in CARA. Further identification, confirmation, and assignment of residues were done in a stripscope mode. The chemical shifts of the assigned residues were used as inputs (*Supplementary file 1t*) in the CS-Rosetta program in the BMRB server (https://csrosetta.bmrb.wisc.edu/csrosetta/submit) to get an ensemble of the 10 lowest energy structures the structure.

Different relaxation rates like $R_1/R_2/R_{1\rho}/R_{2\rho}$ were calculated using mono-exponential decay fitting of peak height against the corresponding set of relaxation delays in a Mathematica script (*Spyracopoulos, 2006*). Steady-state [$^1$H]-$^{15}$N nOes for individual residues were calculated as a ratio of the corresponding residue peak height in the spectra recorded in the presence and absence of $^1$H saturation.

Additional analysis of $^{15}$N-relaxation data ($R_1$, $R_2$, [$^1$H]-$^{15}$N nOe) was conducted using the extended model-free formalism via Relax v4.0.3 software (*Morin et al., 2014*) in a similar fashion carried out previously (*Paithankar et al., 2022*). The structure of TRBP2-dsRBD2 available with PDB ID 2CPN was employed for this analysis.

The effective transverse relaxation rates ($R_{2,eff}$) from CPMG relaxation dispersion experiments, the $R_1$, $R_{1\rho}$, and $R_{2\rho}$ relaxation rates obtained from the HARD experiments, the rotational correlation time, and the subsequent $k_{ex}$, and $p_A$ were calculated by fitting the $R_1$, $R_{1\rho}$, and $R_{2\rho}$ data to a two-state model using numerical fittings as described previously (*Paithankar et al., 2022*; *Spyracopoulos, 2006*). The errors in different fitted parameters were obtained using 500 Monte-Carlo simulations in addition to the duplicate relaxation data points.

The relaxation rates have been depicted in different contexts across the manuscript and the figure legends describe the corresponding context. The raw data have been reported in the Supplementary Information as files.

## MD simulations

All the ensemble cluster analysis was done using UCSF Chimera (*Pettersen et al., 2004*) on the CS-Rosetta (*Lange et al., 2012*; *Shen et al., 2010*) structure of TRBP2-dsRBD2 to select the lowest energy structures. CS-Rosetta structure was used since the previously reported structure (PDB ID: 2CPN) did not have the extended C-terminal sequence used in this study (2CPN has an amino acid length of 150–225, while the construct used in this study is from 154 to 234). Moreover, the buffer conditions of previously reported structures differed from what is used in this study. The lowest energy structures in the ensemble were used as the seed structure for apo TRBP2-dsRBD2 MD simulation. Briefly, GROMACS V.2019.6 (https://www.gromacs.org) was used for all the atomistic MD simulations. All-atom additive CHARMM36 protein force field was used to build the topologies (*Huang et al., 2017*), and initial structure solvation was done TIP3P water molecules in a cubic simulation box with a minimum distance of 1.2 nm from protein in all directions. Microsecond timescale simulations (1 μs) were set up in triplicates in a similar way as described previously (*Paithankar et al., 2022*). All analysis was performed using GROMACS and VMD.

## Acknowledgements

The authors acknowledge NL Prof. Jennifer Doudna (University of California, Berkeley) for the TRBP plasmids. The authors also thank Prof. Gianluigi Veglia (University of Minnesota, Minnesota) and Dr. Fa-An Chao (National Cancer Institute, Bethesda, Maryland) for active discussions while setting up HARD experiments and data analysis. FP acknowledges Dr. Radha Chauhan, Ms. Jyotsana, and Ms. Sangeeta for help with SEC-MALS experiments. The authors acknowledge the High-Field NMR facility at IISER-Pune (co-funded by DST-FIST and IISER Pune). JC acknowledges extramural funding from the Science and Engineering Research Board, Govt. of India (EMR/2015/001966), Department of Biotechnology, Govt. of India (BT/PR24185/BRB/10/1605/2017), and the generous funding from IISER Pune. FP is grateful to DBT-JRF, Govt. of India, for providing a fellowship. ZA and HP thank IISER Pune for the fellowship.

## Additional information

### Funding

| Funder | Grant reference number | Author |
|---|---|---|
| Science and Engineering Research Board | EMR/2015/001966 | Jeetender Chugh |
| Department of Biotechnology, Ministry of Science and Technology, India | BT/PR24185/BRB/10/1605/2017 | Jeetender Chugh |

The funders had no role in study design, data collection, and interpretation, or the decision to submit the work for publication.

### Author contributions

Firdousi Parvez, Conceptualization, Data curation, Formal analysis, Validation, Investigation, Writing – original draft, Writing - review and editing; Devika Sangpal, Data curation, Formal analysis, Investigation, Writing – original draft; Harshad Paithankar, Formal analysis; Zainab Amin, Investigation; Jeetender Chugh, Conceptualization, Supervision, Funding acquisition, Investigation, Writing – original draft, Project administration, Writing - review and editing

### Author ORCIDs

Firdousi Parvez http://orcid.org/0000-0001-8205-0957
Devika Sangpal http://orcid.org/0009-0007-0687-4138
Harshad Paithankar http://orcid.org/0000-0003-0693-9417
Zainab Amin http://orcid.org/0009-0008-5354-1552
Jeetender Chugh https://orcid.org/0000-0002-9996-5202

Reviewer #1 (Public Review): https://doi.org/10.7554/eLife.94842.3.sa1
Reviewer #2 (Public Review): https://doi.org/10.7554/eLife.94842.3.sa2
Author response https://doi.org/10.7554/eLife.94842.3.sa3

## Additional files

### Supplementary files

• Supplementary file 1. Tables providing analyzed data numbers of all the experiments. (**a**) Amino acid length of different secondary structures in TRBP2-dsRBD1 and TRBP2-dsRBD2 CS-ROSSETA structures. α represents an α-helix, L represents a loop, and β represents the β-strand. (**b**) RNA sequences used to study interaction with dsRBDs. (**c**) Nuclear spin relaxation data for apo TRBP2-dsRBD2 recorded at 600 and 800 MHz NMR spectrometer. Data for some residues are missing in this table due to line-broadening issues in the corresponding experiments. (**d**) Order parameter ($S^2$) extracted from model-free analysis of the nuclear spin relaxation data for apo TRBP2-dsRBD2 recorded at 600 and 800 MHz NMR spectrometer. (**e**) $R_{ex}$ of residues extracted from model-free analysis of the nuclear spin relaxation data for apo TRBP2-dsRBD2 recorded at 600 and 800 MHz NMR spectrometer. (**f**) $R_{2,eff}$ values measured at different Carr–Purcell–Meiboom–Gill (CPMG) frequencies from CPMG relaxation dispersion experiment for apo TRBP2-dsRBD2 at 600 MHz NMR spectrometer. (**g**) $R_{1\rho}$ relaxation rates measured using HS$n$ pulses ($n$ = 1, 2, 4, 6, 8) from heteronuclear adiabatic relaxation dispersion (HARD) experiment for apo TRBP2-dsRBD2 at 600 MHz NMR spectrometer. (**h**) $R_{2\rho}$ relaxation rates measured using HSn pulses ($n$ = 1, 2, 4, 6, 8) from HARD experiment for apo TRBP2-dsRBD2 at 600 MHz NMR spectrometer. (**i**) Dynamics parameters extracted from HARD experimental data from geoHARD method for apo TRBP2-dsRBD2. (**j**) Isothermal titration calorimetry (ITC)-binding study of TRBP2-dsRBD2 and D12 RNA carried out in triplicate. (**k**) Nuclear spin relaxation data for RNA-bound TRBP2-dsRBD2 recorded at 600 and 800 MHz NMR spectrometer. Data for some residues are missing in this table due to line-broadening issues in the corresponding experiments. (**l**) Order parameter ($S^2$) extracted from model-free analysis of the nuclear spin relaxation data for RNA-bound TRBP2-dsRBD2 recorded at 600 and 800 MHz NMR spectrometer. (**m**) $R_{ex}$ of residues extracted from model-free analysis of nuclear spin relaxation data for RNA-bound TRBP2-dsRBD2 recorded at 600 and 800 MHz NMR spectrometer. (**n**) $R_{2,eff}$ values measured at different CPMG frequencies from CPMG relaxation dispersion experiment for bound TRBP2-dsRBD2 at 600 MHz NMR spectrometer. (**o**) $R_{1\rho}$ relaxation rates measured using HS$n$ pulses ($n$ = 1, 2, 4, 6, 8) from HARD experiment for RNA-bound TRBP2-dsRBD2 at 600 MHz NMR spectrometer. (**p**) $R_{2\rho}$ relaxation rates measured using HSn pulses ($n$ = 1, 2, 4, 6, 8) from HARD experiment for RNA-bound TRBP2-dsRBD2 at 600 MHz NMR spectrometer. (**q**) Dynamics parameters extracted from HARD experimental data from geoHARD method for RNA-bound TRBP2-dsRBD2. (**r**) $R_1$ relaxation rates measured from HARD experiment for apo TRBP2-dsRBD2 at 600 MHz NMR spectrometer. (**s**) $R_1$ relaxation rates measured from HARD experiment for RNA-bound TRBP2-dsRBD2 at 600 MHz NMR spectrometer. (**t**) Assignment Report of TRBP2-dsRBD2 (as obtained from CARA in NMR-STAR 3.1 format).

• MDAR checklist

### Data availability

All the analyzed NMR data has been included as supplementary files. All the raw data (NMR data in Bruker format, ITC data, SEC-MALS data) has been uploaded to the Mendeley server as per figure numbers and is available to download at https://doi.org/10.17632/cgyb8tv7n7.1.

The following dataset was generated:

| Author(s) | Year | Dataset title | Dataset URL | Database and Identifier |
|---|---|---|---|---|
| Parvez F, Sangpal D, Paithankar H, Amin Z, Chugh J | 2024 | Differential conformational dynamics in two type-A RNA-binding domains drive the double-stranded RNA recognition and binding | https://doi.org/10.17632/cgyb8tv7n7.1 | Mendeley Data, 10.17632/cgyb8tv7n7.1 |

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
