## [Editor Report · eLife assessment]

This study presents a **useful** comparison of the dynamic properties of two RNA-binding domains. The data collection and analysis are **solid**, making excellent use of a suite of NMR experiments and ITC data. Nonetheless, reported evidence was found to only partially support the proposed connection between the backbone dynamics of the tandem domains and their RNA binding activity. This work will be of interest to biophysicists working on RNA-binding proteins.

---

## [Referee Report · Reviewer #1 (Public Review)]

In the manuscript Chugh and co-workers utilize a suite of NMR relaxation methods to probe the dynamic landscape of the TAR RNA binding protein (TRBP) double-stranded RNA-binding domain 2 (dsRBD2) and compare these to their previously published results on TRBP dsRBD1. The authors show that, unlike dsRBD1, dsRBD2 is a rigid protein with minimal ps-ns or us-ms time scale dynamics in the absence of RNA. They then show that dsRBD2 binds to canonical A-form dsRNA with a higher affinity and with less changes in dynamics compared to dsRBD1.

Strengths:

The authors expertly use a variety of NMR techniques to probe protein motions over six-orders of magnitude in time. Other NMR titration experiments and ITC data support the RNA-binding model.

Weaknesses:

Generally, the data collection and analysis are sound. However, microsecond timescale dynamics for the RNA-bound form of dsRBD2 are inferred from a sample that is only 5% bound. Additionally, the manuscript lacks context with the much broader field of RNA-binding proteins. For example, many studies have shown that RNA recognition motif (RRM) domains have similar dynamic characteristics when binding diverse RNA substrates.

---

## [Referee Report · Reviewer #2 (Public Review)]

Summary:

Proteins that bind to double-stranded RNA regulate various cellular processes, including gene expression and viral recognition. Such proteins often contain multiple double-stranded RNA-binding domains (dsRBDs) that play an important role in target search and recognition. In this work, Chug and colleagues have characterized the backbone dynamics of one of the dsRBDs of a protein called TRBP2, which carries two tandem dsRBDs. Using solution NMR spectroscopy, the authors characterize the backbone motions of dsRBD2 in the absence and presence of dsRNA and compare these with their previously published results on dsRBD1. The authors show that dsRBD2 is comparatively more rigid than dsRBD1 and claim that these differences in backbone motions are important for target recognition.

Strengths:

The strengths of this study are multiple solution NMR measurements to characterize the backbone motions of dsRBD2. These include 15N-R1, R2, and HetNOE experiments in the absence and presence of RNA and the analysis of these data using an extended-model-free approach; HARD-15N-experiments and their analysis to characterize the kex. The authors also report differences in binding affinities of dsRBD1 and dsRBD2 using ITC and have performed MD simulations to probe the differential flexibility of these two domains.

Weaknesses:

While it may be true that dsRBD2 is more rigid than dsRBD1, the manuscript lacks conclusive and decisive proof that such changes in backbone dynamics are responsible for target search and recognition and for the diffusion of TRBP2 along the RNA molecule.

---

## [Author Response]

The following is the authors’ response to the original reviews.

**eLife assessment**
This study presents a useful comparison of the dynamic properties of two RNA-binding domains. The data collection and analysis are solid, making excellent use of a suite of NMR methods. However, evidence to support the proposed model linking dynamic behavior to RNA recognition and binding by the tandem domains remains incomplete. The work will be of interest to biophysicists working on RNA-binding proteins.

We thank eLife for taking the time and effort to review our manuscript. Evidence from the literature and our study shows a great deal of parity between the dynamic behavior of dsRBDs and its dsRNA-recognition and -binding that helped us culminate in proposing a fair model. As already mentioned in the manuscript, we have been working on the suggested experiments to support our proposed model further.

**Public Reviews**:**Reviewer #1** (Public Review):Summary:In the manuscript entitled "Differential conformational dynamics in two type-A RNA-binding domains drive the double-stranded RNA recognition and binding," Chugh and co-workers utilize a suite of NMR relaxation methods to probe the dynamic landscape of the TAR RNA binding protein (TRBP) double-stranded RNA-binding domain 2 (dsRBD2) and compare these to their previously published results on TRBP dsRBD1. The authors show that, unlike dsRBD1, dsRBD2 is a rigid protein with minimal ps-ns or us-ms time scale dynamics in the absence of RNA. They then show that dsRBD2 binds to canonical A-form dsRNA with a higher affinity compared to dsRBD1 and does so without much alteration in protein dynamics. Using their previously published data, the authors propose a model whereby dsRBD2 recognizes dsRNA first and brings dsRBD1 into proximity to search for RNA bulge and internal loop structures.

We thank the Reviewer for sending us an encouraging review. We have combined the findings reported in the literature with new ones that led us to propose the dsRNA-binding model by tandem A-form dsRBDs.

We propose that dsRBD1 can first recognize a variety of sequential and structurally different dsRNAs. dsRBD2 assists the interaction with a higher affinity, thus fortifying the interaction between TRBP and a possible substrate. This may enable the other associated proteins like Dicer and Ago2 to perform critical biological functions.

However, we feel that a few statements in the comment above are factually incorrect.

Statement 1. “They then show that dsRBD2 binds to canonical A-form dsRNA with a higher affinity compared to dsRBD1 and does so without much alteration in protein dynamics.”

We have explicitly shown the perturbation in dsRBD2 dynamics upon RNA binding.

Statement 2. “Using their previously published data, the authors propose a model whereby dsRBD2 recognizes dsRNA first and brings dsRBD1 into proximity to search for RNA bulge and internal loop structures.”

Our previously published data suggests that dsRBD1, owing to its high conformational dynamics in solution, is able to recognize a variety of structurally and sequentially different dsRNAs ([Paithankar et al., 2022]). dsRBDs preferably bind to the double-stranded region (minor-major-minor-groove) of an A-form RNA ([Acevedo et al., 2016]; [Vuković et al., 2014]) and do not search for bulge and internal loop structures as a part of the binding event. Even though dsRBDs preferably bind to the double-stranded region, they can still accommodate perturbation in the A-form helix due to mismatch and bulges with decreased binding affinity ([Acevedo et al., 2015]). However, it is a matter of future research to identify how much of a deviation from the A-form structure can be accommodated by the dsRBDs. The diffusion event observed in the literature ([Koh et al., 2013]) also does not show any direct implication for searching for bulge and internal loop structures.

Strengths:The authors expertly use a variety of NMR techniques to probe protein motions over six orders of magnitude in time. Other NMR titration experiments and ITC data support the RNA-binding model.Weaknesses:The data collection and analysis are sound. The only weakness in the manuscript is the lack of context with the much broader field of RNA-binding proteins. For example, many studies have shown that RNA recognition motif (RRM) domains have similar dynamic characteristics when binding diverse RNA substrates. Furthermore, there was no discussion about the entropy of binding derived from ITC. It might be interesting to compare with dynamics from NMR.

We understand the reviewer’s point that this study is focused on a dsRNA-binding mechanism rather than addressing the much broader field of RNA-binding. There are multiple challenges in finding a single mechanism that works for all RNA-binding proteins. For instance, RRM is a single-stranded RNA binding domain that is able to read out the substrate base sequence. RRM behaves entirely differently than the dsRBD in terms of target specificity. Besides, several other RNA-binding domains, like the KH-domain, Puf domains, Zinc finger domains, etc., showcase a unique RNA-binding behavior. Thus, it would be really difficult to draw a single rule of thumb for RNA-recognition behavior for all these diverse domains.

Thank you for pointing out the entropy of binding from ITC. We have now included the entropy of binding discussion in the main text, page 7.

**Reviewer #2 (Public Review):**
Summary:Proteins that bind to double-stranded RNA regulate various cellular processes, including gene expression and viral recognition. Such proteins often contain multiple double-stranded RNA-binding domains (dsRBDs) that play an important role in target search and recognition. In this work, Chug and colleagues have characterized the backbone dynamics of one of the dsRBDs of a protein called TRBP2, which carries two tandem dsRBDs. Using solution NMR spectroscopy, the authors characterize the backbone motions of dsRBD2 in the absence and presence of dsRNA and compare these with their previously published results on dsRBD1. The authors show that dsRBD2 is comparatively more rigid than dsRBD1 and claim that these differences in backbone motions are important for target recognition.Strengths:The strengths of this study are multiple solution NMR measurements to characterize the backbone motions of dsRBD2. These include 15N-R1, R2, and HetNOE experiments in the absence and presence of RNA and the analysis of these data using an extended-model-free approach; HARD-15N-experiments and their analysis to characterize the kex. The authors also report differences in binding affinities of dsRBD1 and dsRBD2 using ITC and have performed MD simulations to probe the differential flexibility of these two domains.Weaknesses:While it may be true that dsRBD2 is more rigid than dsRBD1, the manuscript lacks conclusive and decisive proof that such changes in backbone dynamics are responsible for target search and recognition and the diffusion of TRBP2 along the RNA molecule. To conclusively prove the central claim of this manuscript, the authors could have considered a larger construct that carries both RBDs. With such a construct, authors can probe the characteristics of these two tandem domains (e.g., semi-independent tumbling) and their interactions with the RNA. Additionally, mutational experiments may be carried out where specific residues are altered to change the conformational dynamics of these two domains. The corresponding changes in interactions with RNA will provide additional evidence for the model presented in Figure 8 of the manuscript. Finally, there are inconsistencies in the reported data between different figures and tables.

We thank the reviewer for the comprehensive and insightful review. A larger construct carrying both RBDs was not used because of the multiple challenges pertaining to dynamics study by NMR spectroscopy (intrinsic R2 rates of the dsRBD1-dsRBD2 construct would be high, resulting in broadened peaks) as per our previous experience ([Paithankar et al., 2022]). There would be additional dynamics in that construct coming from domain-domain relative motions, and it is difficult to deconvolute the dynamics information. Further, the dsRNA needed to bind to this construct will be longer, causing further line broadening in NMR.

Coming to mutational studies, careful designing of domain mutants remains as a challenge because the conformational dynamics in both the domains are distributed all through the backbone rather than only in the RNA-binding residues. The mutational studies would need an exhaustive number of mutations in protein as well as RNA to draw a parallel between the binding and dynamics. Having said that, we are working on making such mutations in the protein (at several locations to freeze the dynamics site-specifically) and the RNA (to change the shape of the dsRNA) to systematically study this mechanism, which will be out of scope of this manuscript.

The reviewer has rightly pointed out some subtle superficial differences in the reported data between different figures and tables. These superficial differences are present because of the context in which we are describing the data. For example, in Figure S4, we are talking about the average relaxation rates and nOe values for only the common residues we were able to analyze between two magnetic field strengths 600 and 800 MHz. Whereas in Figure 6, we are comparing the averages of the core (159-227) dsRBD residues at 600 MHz, in the presence and absence of D12RNA. The differences, however, are minute falls well within the error range.

**Recommendations for the authors:**

**Reviewer #1 (Recommendations For The Authors):**
Suggestions for improved or additional experiments -In regards to ITC data, dsRBD1 does not bind canonical A-form RNA with high affinity. What is dsRBD1 and dsRBD2 affinity to the miR-16 RNA?

We have not performed ITC-based studies with miR-16 RNA for the domains. The study by Acevedo et al. has shown the effect of lengths of Watson-Crick duplex RNAs upon TRBP2 dsRBD binding. In this study, they have compared the ds22 RNA to miRNA/miRNA* duplex. By using EMSA, they show that the Kd,app (μM) for dsRBD1 is 3.5±0.2 and for dsRBD2 is 1.7±0.1, indicating a higher affinity by the latter ([Acevedo et al., 2015]).

What was the amount of time used for the 1H saturation in the heteronuclear NOE experiment? Based on the average T1 (1/1.44 s-1) = 0.69 s, a recovery delay of >7 s should have been used for this experiment.

According to Cavanagh et al., a minimum recovery/recycle delay should be greater than 5*1/R1 to make sure that 99% of the 1HN and 15N magnetizations are restored ([“Protein NMR Spectroscopy, Principles and Practice, John Cavanagh, Wayne J. Fairbrother, Arthur G. Palmer III, and Nicholas J. Skelton. Academic Press, San Diego, 1995, 587 pages, $59.95. ISBN: 0-12-164490-1.,” 1996]). In our study, we have used a relaxation delay of 5 s, which is greater than 7*1/R1avg thus ensuring at least 99% of the 1HN and 15N recover their bulk magnetization.

Recommendations for improving writing and presentation -Figure 3 - The legend in panel C is incomplete.

Figure 3 (Figure 4 in the revised manuscript) has been updated, and the legend now reads complete.

Figures 3 E and F - The three views can be combined into one as is done in Figures 4 C and D.

Thanks for the kind suggestion. We have depicted the kex in the three ranges to highlight the difference between the two domains at each range. Since there are three different exchange regimes with different populations, we believe this gives us an uncomplicated picture while classifying and comparing the dynamics between the two. Combining the three views into one becomes too overwhelming to visualize kex and population distribution in the protein.

Figure 3 - The residues indicated in the text (e.g., R200, L212, and R224) should be indicated in panels E and F.

We have marked the residues described in the text in Figure 4C (revised Figure 5C), and thus, they are not mentioned in Figures 3E and 3F (revised Figures 4E and 4F).

The results and discussion put these findings into minimal context. Most comparisons are made between dsRBD1 and dsRBD2. What about other RNA-binding proteins? There is a wealth of structure/dynamics/functional data about RNA recognition motifs, which do exactly the same thing as described here but are missing.

We understand the reviewer’s point that this study is focused on a dsRNA-binding mechanism rather than addressing the much broader field of RNA-binding. There are multiple challenges in finding a single mechanism that works for all RNA-binding proteins. For instance, RRM is a single-stranded RNA-recognition motif that can read out the substrate base sequence. RRM behaves entirely differently than the dsRBD in terms of sequence specificity. Besides, several other RNA-binding domains, like the KH-domain, Puf domains, Zinc-finger domains, etc., showcase a unique RNA-binding behaviour. Thus, with the current knowledge, it would not be possible to draw a single rule of thumb for RNA-recognition behaviour for all these diverse domains. Hence, the findings of this study are not comparable to those of other RNA-binding domains and are beyond the scope of this study.

Results, page 8 - I'm not sure that allosteric quenching is appropriately invoked here. The amount of residues showing dynamics in the apo state is small and the number only moderately increases upon RNA binding. The observation that some residues show an increase and a neighboring residue shows a decrease (or vice versa) upon RNA binding could just be random with the small number of observations. This observation would be more convincing if it were happening to larger regions within the protein.

We agree with the reviewer that the number of residues showing dynamics in the apo-state of the dsRBD2 is small when compared with that of dsRBD1, and the number only moderately increases upon RNA-binding. However, we believe it is quite important to invoke the allosteric quenching as all the new residues where dynamics is induced, do lie in the spatial proximity, as also observed in the dsRBD1 ([Paithankar et al., 2022]). It is a parameter to not only compare the differences and similarities in the two domains but also to highlight the presence of this phenomenon common in both the type-A dsRBDs of TRBP.

Minor corrections -Introduction, page 2 - The order parameter should be defined for non-NMR experts.

Thank you for the suggestion. The definition of order parameter has now been included on page 2 of the revised manuscript.

Introduction, page 2 - TRBP should be defined in the main text the first time used.

We have now defined TRBP on page 2 of the revised manuscript, where it is used in the main text for the first time.

Results, page 5 - The reference for the HARD experiment should be given earlier in that paragraph.

Thank you for the suggestion. We have now referenced the HARD experiment earlier in the last paragraph on page 5 of the revised manuscript.

Results, page 7 - What is the limiting amount of RNA used for the D12-bound dsRBD2 spin relaxation measurements?

The limiting amount of RNA used for the D12-bound dsRBD2 spin relaxation measurements is 0.05 equivalent (RNA:Protein = 50 mM:1000 mM). It has now been included on page 7 of the revised manuscript.

**Reviewer #2 (Recommendations For The Authors):**
Throughout the manuscript, NMR datasets are not consistent with one another (a few examples are listed below).Figures S4, 6, and Table S4: (a) It is unclear why relaxation data for certain residues are missing in Table S4 (e.g., S156, V168, E177, F192, etc.).

We thank the reviewer for pointing this out. We have now reanalyzed the data for all the above-mentioned residues and other missing residues. In the revised manuscript, we have added the data for the above-mentioned residues like E177, R189, and many more N- and C-terminal residues. Unfortunately, for some residues like V168, S184, F192, S209, and L222, we witnessed severe peak broadening while measuring the R2 rates and/or nOe. Hence, data for V168, S184, F192, S209, and L222 are missing in Table S4. We have explicitly mentioned this in the table legends about missing data for a few residues.

(b) The reported values are not consistent. For example, Figure S4 says that the average 15N-R2 rate is 10.85 +/- 0.36 s-1 whereas Figure 6 says the 15N-R2 rate is 11.02 +/- 0.39 s-1 for the same dataset.

The superficial differences are present because of the context in which we are describing the data (now mentioned in the methods section on page 13). In Figure S4, we are talking about the average relaxation rates and nOe values for only the common residues we could analyze between two magnetic field strengths, 600 and 800 MHz. Whereas in Figure 6 (revised figure 3), we compare the averages of all the analyzed core dsRBD residues at 600 MHz in the presence and absence of D12RNA. The differences, however, are insignificant, falling well within the error range.

(c) There is also a discrepancy in reported R2 values (at 600 MHz) in Table S4. It is unclear to me what the reported values are, as most of these are below 1 s-1.

Thank you very much for pointing out our mistake here. The Table S4 seems to have the wrong values for R2 at 600 MHz. However, the raw data submitted to the BMRB as entry 52077 holds the correct information. We have now updated the Table S4.

(d) It is also unclear as to why perfectly resolved residues (e.g., L230, A232, D234, etc.) have been omitted from these data (and other datasets such as 15N-CPMGs shown in Figure S6).

The residues L230, A232, D234, etc., are the C-terminal residues of TRBP-dsRBD2 beyond the core (159-227 aa) fold of dsRBD. They have now been included in the revised figures S6 and S11 for completeness.

(e) Figure 6 reports a 15N-R2 of 21 s-1 for one of the residues in the absence of RNA. This data point has been omitted from Figure S4.

In Figure S4, we are talking about relaxation rates and nOe values only for the common residues we could analyze between the two magnetic field strengths, 600 and 800 MHz. Thus, that 15N-R2 value has been omitted.

The S2 order parameters reported in Figures S5 and S10 are inconsistent with one another, as additional residues are shown in S10 (e.g., N159).

Thank you for pointing it out. We have now reanalyzed the data for S2 order parameter and Rex by including more residues (e.g., N159, R189, etc) in the core and have updated both Figures S5 and S10. Please see the revised supplementary information.

Tables S6 and S7 report values for residue R189. This residue has been omitted in every other dataset. Based on the 1H-15N HSQC spectrum shown in Figure S3, this residue gives a well-resolved crosspeak (which lies adjacent to V228). Can the authors explain why they omit data for this residue in Figures S4, 6, and Table S4?

The reviewer is correct in pointing out that data for R189 is missing in the fast dynamics data, such as Figure S4, Figure 6 (revised figure 3), and Table S4. We have now reanalyzed our raw data and included data for R189 and other missing residues in our updated manuscript. Please see the revised figures S4 and 6 (revised figure 3) and the revised table S4.

Moreover, this residue lies in the loop2 region of this domain. Based on the MD simulations (Figure 2), this region is more flexible compared to the rest of the domain. Does the corresponding 15N-relaxation data support this claim?

Yes, the apo 15N-relaxation data do strongly support this claim. R189 showed a higher than core average R2 rate (R189 = 15.44 +/- 0.69 s-1; core = 10.92 +/- 0.37 s-1) and a lower than core average nOe (R189 = 0.49 +/- 0.05; core = 0.73 +/- 0.03) which indicate a higher flexibility than the rest of the core (updated Figure 3 and Table S4). Additionally, the S2 order parameter for R189 was found to be 0.52 +/- 0.03, slightly lower than the core average of 0.59 +/- 0.03, indicating a more flexible region than the core (updated Table S14). Moreover, the dynamics parameters extracted from HARD experimental data using the geoHARD method for apo TRBP2-dsRBD2 shown in Table S18 depict a high kex value of 31748.72 +/- 955.20 Hz for R189. This supports the claim that this residue is highly flexible with a high exchange rate.

Figure S9. I was not able to follow this dataset as the data points are not consistent between different residues.

In Figure S9, the residue-wise peak intensities plotted against the RNA concentration indicate that line broadening was witnessed for all the core residues (irrespective of the initial peak intensity). Another interesting observation is that the terminal residues do not undergo the same line broadening as seen in the core residues.

It is also unclear why residue G185 is highlighted.

It is taken as an example and magnified to show the extent of line broadening. This is now explicitly mentioned in the figure caption in the revised supplementary information.

It is also not clear exactly what the authors are trying to fit, as I see no chemical shift changes upon the addition of RNA (Fig. S8), and the equation used for data fitting (pg. 11) uses chemical shift changes (and not the changes in intensities).

The same equation can be used to fit the chemical shift perturbation and peak intensity perturbation as a function of ligand concentration. Here, we have tried to fit the intensity perturbation. We have now modified the statement on page 11 in the revised manuscript.

Table S2: The ITC analysis reports an n value of ~3. Can authors elaborate as to what this means?

The stoichiometry ~3 indicates the number of TBDP2-dsRBD2 that can interact with D12 RNA in a single binding event. The minimum binding register for dsRBDs is known to be >8 bp (12 bp for optimal binding) ([Ramos et al., 2000]), and one single domain only covers one-third of the face of the cylindrical RNA ([Masliah et al., 2018]). Hence, 3 dsRBD2 could interact with a 12-mer RNA in solution.

The reported Kd values between the main text (page 7) and Figure 5 are not consistent with one another (one lists 1.18 uM while the other says 1.11 uM). Table S2 does not list the parameters for interactions between dsRBD1 and D12.

Figure 5 (revised figure 6) depicts the information of a single isolated experiment out of a total of three, whereas in the main text, we say 1.18 μM as the average Kd value (table S2).

Figure S4: The red axis should read "211" instead of "111".

Thank you for your helpful insight. We have now changed it in the revised figure.

Table S3 lists the structural motifs of the two dsRBDs, which are nearly identical to one another, and yet the manuscript claims that these are different (page 4, paragraph 1).

We agree with the reviewer that the differences are minute but important, which we have tried to highlight in this paper. In particular, loop 2, critical for dsRNA-binding ([Masliah et al., 2012]), is 1 residue longer in dsRBD2 and has a possible effect in enhanced substrate binding.

Figure S8 shows severe signal attenuation for many residues upon the addition of 100 uM RNA. The most notable among these are residues M194, T195, and C196. Can the authors explain how they measure 15N-relaxation rates for these residues in the presence of 50 uM D12?

First, we have recorded the measured 15N-relaxation rates for these residues in the presence of 50 mM D12 (RNA:Protein = 50 mM:1000 mM), corresponding to 0.05 equivalent RNA. The amount of RNA used is less than that used for the HSQC-based titration shown in Figure S8, 0.1 equivalent RNA (RNA:Protein = 5 mM:50 mM), where we witness line broadening for residues like M194, T195, and C196. Second, we increased the overall protein concentration from 50 mM (used in HSQC-based titration) to 1000 mM (used in relaxation measurements) to ensure a better signal-to-noise ratio in all the spectra.

Use the same coloring scheme for Figures S7 and S8.

Thank you for the suggestion. We have now edited Figure S8 accordingly.

Figures are often listed out-of-order, making it difficult to follow the manuscript.

Thank you for the suggestion. We have now amended the main text to refer to the figures sequentially. While doing so, we have renumbered Figure 6 as Figure 3, Figure 3 as Figure 4, Figure 4 as Figure 5, and Figure 5 as Figure 6.

Figure captions for the relaxation data should specify the temperature at which these datasets were collected.

Thanks for the valuable suggestion. We have now added the temperature wherever applicable.

References

Acevedo R, Evans D, Penrod KA, Showalter SA. 2016. Binding by TRBP-dsRBD2 Does Not Induce Bending of Double-Stranded RNA. Biophys J 110:2610–2617. doi:10.1016/j.bpj.2016.05.012

Acevedo R, Orench-Rivera N, Quarles KA, Showalter SA. 2015. Helical Defects in MicroRNA Influence Protein Binding by TAR RNA Binding Protein. PLoS ONE 10:e0116749. doi:10.1371/journal.pone.0116749

Koh HR, Kidwell MA, Ragunathan K, Doudna JA, Myong S. 2013. ATP-independent diffusion of double-stranded RNA binding proteins.

Masliah G, Barraud P, Allain FH-T. 2012. RNA recognition by double-stranded RNA binding domains: a matter of shape and sequence. Cell Mol Life Sci 70:1875–1895. doi:10.1007/s00018-012-1119-x

Masliah G, Maris C, König SL, Yulikov M, Aeschimann F, Malinowska AL, Mabille J, Weiler J, Holla A, Hunziker J, Meisner‐Kober N, Schuler B, Jeschke G, Allain FH. 2018. Structural basis of siRNA recognition by TRBP double‐stranded RNA binding domains. EMBO J 37:e97089. doi:10.15252/embj.201797089

Paithankar H, Tarang GS, Parvez F, Marathe A, Joshi M, Chugh J. 2022. Inherent conformational plasticity in dsRBDs enables interaction with topologically distinct RNAs. Biophys J 121:1038–1055. doi:10.1016/j.bpj.2022.02.005

Protein NMR Spectroscopy, Principles and Practice, John Cavanagh, Wayne J. Fairbrother, Arthur G. Palmer III, and Nicholas J. Skelton. Academic Press, San Diego, 1995, 587 pages, $59.95. ISBN: 0-12-164490-1. 1996. . J Magn Reson, Ser B 113:277. doi:10.1006/jmrb.1996.0189

Ramos A, Grünert S, Adams J, Micklem DR, Proctor MR, Freund S, Bycroft M, Johnston DS, Varani G. 2000. RNA recognition by a Staufen double‐stranded RNA‐binding domain. EMBO J 19:997–1009. doi:10.1093/emboj/19.5.997

Vuković L, Koh HR, Myong S, Schulten K. 2014. Substrate Recognition and Specificity of Double-Stranded RNA Binding Proteins. Biochemistry 53:3457–3466. doi:10.1021/bi500352s